

# Circulation of the Turkish Straits System between 2008-2013 under complete atmospheric forcings

Ali Aydoğdu[1,2,3], Nadia Pinardi[2,4], Emin Özsoy[5,6], Gokhan Danabasoglu[7], Özgür Gürses[5,8], and Alicia Karspeck[7]

[1]Science and Management of Climate Change, Ca' Foscari University of Venice, Italy
[2]Centro Euro-Mediterraneo sui Cambiamenti Climatici, Bologna, Italy
[3]Nansen Environmental and Remote Sensing Center, Bergen, Norway
[4]Department of Physics and Astronomy, University of Bologna, Italy
[5]Institute of Marine Sciences, Middle East Technical University, Erdemli, Turkey
[6]Eurasia Institute of Earth Sciences, Istanbul Technical University, Turkey
[7]National Center for Atmospheric Research, Boulder, Colorado
[8]Alfred Wegener Institute for Polar and Marine Research, Bremerhaven, Germany

**Correspondence:** Ali Aydoğdu (ali.aydogdu@nersc.no)

**Abstract.** A simulation of the Turkish Straits System (TSS) using a high-resolution, three-dimensional, unstructured mesh ocean circulation model with realistic atmospheric forcing for the 2008-2013 period is presented. The depth of the interface layer between the upper and lower layers remains stationary after six years of integration, indicating that despite the limitations of the modelling system, the simulation maintains its realism. The solutions capture important responses to high frequency atmospheric events such as the reversal of the upper layer flow in the Bosphorus due to southerly severe storms, i.e., blocking events, to the extent that such storms are present in the forcing dataset. The annual average circulations show two distinct patterns in the Marmara Sea. When the wind stress maximum is localised in the central basin, the Bosphorus jet flows to the south and turns west after reaching the Bozburun Peninsula. In contrast, when the wind stress maximum increases and expands in the north-south direction, the jet deviates to the west before reaching the southern coast and forms a cyclonic gyre in the central basin. In certain years the mean kinetic energy in the northern Marmara Sea is found to be comparable to that of the Bosphorus inflow.

## 1 Introduction

The Turkish Straits System (TSS) connects the Marmara, Black and Mediterranean Seas through the Bosphorus and Dardanelles Straits. The near-surface layer of low salinity waters originating from the Black Sea enters from the Bosphorus, flowing west and exiting into the Aegean Sea at the Dardanelles. The deeper, more saline waters of Mediterranean origin enter from the Dardanelles in the lower layer and eventually reach the Black Sea through the undercurrent of the Bosphorus. The strongly stratified marine environment of the TSS is characterised by a sharp pycnocline positioned at a depth of 25 m (Ünlüata et al., 1990). The complex topography of the Marmara Sea consists of a wide shelf in the south, a narrower one along the north-





ern coast and three east-west deep basins separated by sills, connected to the two shallow, elongated narrow straits providing passage to the adjacent seas at the two ends, as shown in Fig. 1.

The TSS mass and property balances are mainly controlled by the Black Sea in the upstream. At the Bosphorus Black Sea entrance, the long-term salinity budget implies a ratio of about two between the upper and lower layer volume fluxes (Peneva
et al., 2001; Kara et al., 2008). The net flux is estimated to be comparable to the Black Sea river runoff, as the annual average precipitation and evaporation over the sea surface are roughly of the same order (Özsoy and Ünlüata, 1997). Daily to seasonal variations in net fluxes through the TSS are driven by changes in Black Sea river runoffs, barometric pressure and wind forcing.

Climatological means of water and tracer fluxes through the TSS were initially estimated from long-term observations of seawater properties at junctions of the straits and on surface water fluxes (Ünlüata et al., 1990; Beşiktepe et al., 1994; Tuğrul
et al., 2002; Maderich et al., 2015), followed later by ship-borne and moored ADCP measurements at the straits (Özsoy et al., 1988; Özsoy et al., 1998; Altıok et al., 2012; Jarosz et al., 2011b, a, 2012, 2013). Updated reviews of TSS fluxes based on combined data have been provided by Schroeder et al. (2012); Özsoy and Altıok (2016); Sannino et al. (2017); Jordá et al. (2017).

The hydrodynamic processes of the TSS extend over a wide range of interacting space and time scales. The complex to-
pography of the straits and property distributions have resulted in hydraulic controls being anticipated in both straits (Özsoy et al., 1998; Özsoy et al., 2001), which can only partially be demonstrated by measurements at the northern sill of the Bosphorus (Gregg and Özsoy, 2002; Dorrell et al., 2016). Hydraulic controls have since been found by modelling at the southern contraction-sill complex and the northern sill, confirming a unique maximal exchange regime adjusted to the particular topography and stratification (Sözer and Özsoy, 2017a; Sannino et al., 2017). These findings support the notion that the Bosphorus
is the more restrictive of the two straits in controlling the outflow from the Black Sea to the Mediterranean. The analysis of moored measurements by Book et al. (2014) demonstrated this, and indicated a more restrained sea level response transmitted across the Bosphorus than in the Dardanelles.

Improvements in modelling have provided a better scientific understanding of the TSS circulation, and they can now address the complex processes characterising the system. The initial step in this formidable task is to construct separate models of
the individual compartments of the system, which are the two Straits and the Marmara basin. The first simplified models of the Bosphorus were by Johns and Oğuz (1989) who solved the turbulent transport equations in 2D, and found a two-layer stratification to develop. Simplified two-layer or laterally averaged models of the Dardanelles and Bosphorus were later developed by Oğuz and Sur (1989), Stashchuk and Hutter (2001) and Oğuz et al. (1990) respectively, while Hüsrevoğlu (1998) introduced a 2D reduced gravity ocean model of the Dardanelles inflow into the Marmara Sea. Similar 2D laterally averaged
models (Maderich and Konstantinov, 2002; Ilıcak et al., 2009; Maderich et al., 2015) and 3D models (Kanarska and Maderich, 2008; Öztürk et al., 2012), which were of limited extent, have been used to construct simplified solutions for the Bosphorus exchange flows. Bosphorus hydrodynamics were extensively investigated by Sözer and Özsoy (2017a) using a 3D model with turbulence parameterisation under idealised and realistic topography with stratified boundary conditions in adjacent basins, demonstrating the unique hydraulic controls in the maximal exchange regime that are to be established in the realistic case.
The combined effects of the Bosphorus and the proposed parallel channel known as Kanal İstanbul have been investigated by



Sözer and Özsoy (2017b), indicating weak coupling between the two channels, which have very different characteristics, but this has been found to be of climatic significance in modifying the fluxes across the TSS.

Very few studies have attempted to model the circulation in the Marmara Sea, even as a stand-alone system excluding the dynamical influences of the straits and atmospheric forcing. Chiggiato et al. (2012) modelled the Marmara Sea using realistic atmospheric forcing and open boundaries at the junctions of the straits with the Sea, indicating surface circulation changes in response to changes in the strength and directional pattern of the wind force.

Similarly, the interannual variability of the Marmara Sea has been examined by Demyshev et al. (2012) using open boundary conditions at the strait junctions in the absence of atmospheric forcing. They reproduced the S-shaped jet current traversing the basin under the isolated conditions of a net barotropic current, which with appropriate parameterisation successfully preserved the sharp interface between the upper and lower layers when the model steady-state was reached after 18 years of simulation. The S-shaped upper layer circulation of the Marmara Sea predicted by Demyshev et al. (2012) appears similar to what Beşiktepe et al. (1994) found in summer, when wind forcing is at its minimum or at least close to being in a steady-state. An anti-cyclonic pattern has generally been identified in the central Marmara Sea, like the cases reported by Beşiktepe et al. (1994).

The challenges of modelling the entire TSS domain were recently undertaken by Gürses et al. (2016). The effects of atmospheric forcing were considered, excluding the effects of the net flux through the TSS. The study used an unstructured triangular mesh model, the Finite Element Sea-Ice Ocean Model (FESOM), with a high horizontal resolution reaching about 65 m in the straits in the horizontal. The water column is discretized by 110 vertical levels.

The study of Sannino et al. (2017) used curvilinear coordinate implementation of the MITgcm (Marshall et al., 1997) with a non-uniform grid in the horizontal, a minimum of 65 m resolution in the narrowest part of Bosphorus and 100 levels in the vertical. The model was used to investigate the circulation of the TSS under varying barotropic flow through the system in the absence of atmospheric forcing. The overall circulation in the Marmara Sea was found to differ significantly with variations in the net volume transport at the Bosphorus. The circulation changes from a large anticyclonic circulation at the centre of the basin at low flux values, to different gyres wrapped around an S-shaped jet as the net flux is increased. A cyclonic central gyre is eventually generated as the increased flux leads to the lower layer in the Bosphorus being blocked. The most significant finding was the nonlinear sea level response, which deviated widely from the linear response predicted by the stand-alone Bosphorus model of Sözer and Özsoy (2017a).

Stanev et al. (2017) approached the challenge by using an unstructured mesh model. The model covers the entire Black Sea, and is seamlessly linked to the TSS and the northern Aegean Sea with open boundaries in the south and uses realistic atmospheric forcing. The focus is on the transport at the straits and the resulting dynamics of the Black Sea. Regarding the TSS, the results support the notion of multiple controls of the barotropic flow. They were also able to simulate short-term events such as severe storm passages. However, in a high-resolution model of the TSS, the Courant-Friedrichs-Lewy (CFL) condition (Courant et al., 1967) restricts the time step to a few seconds with explicit schemes. As a solution, Stanev et al. (2017) used an implicit advection scheme for transport to handle a wide range of Courant numbers (Zhang et al., 2016) while satisfying the stability of the solution. However, the computational burden of using an implicit scheme imposed a coarser model resolution



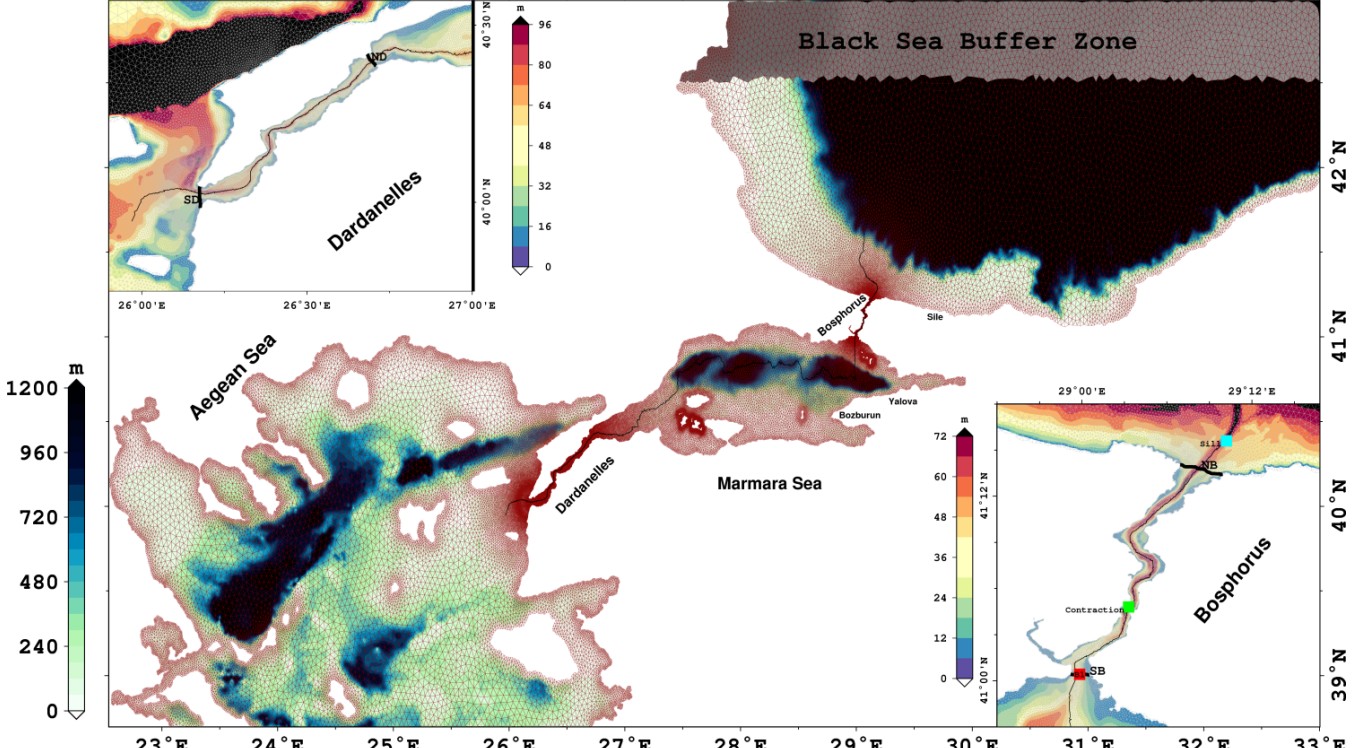

**Figure 1.** Bathymetry of the Turkish Straits System and the model domain. Bathymetry of the Bosphorus and Dardanelles Straits are detailed in the small panels. The colours represent the depth. The colourbar scales are different for the straits and the whole domain. A triangular mesh is overlaid with red for the entire domain and with grey for the straits in small panels. The Thalweg used to display the cross-section throughout the TSS is represented by the black line. The grey shaded area in the Black Sea functions as a buffer zone and is described in the text. Cross-sections at the boundaries of the straits are used for volume flux computations. NB, SB, ND and SD in the small panels are northern Bosphorus, southern Bosphorus, northern Dardanelles and southern Dardanelles, respectively. In the Bosphorus panel, the green and cyan squares show the locations of the contraction and northern sill, respectively. Finally, the red square B1 indicates the middle of the Marmara Sea exit of the Bosphorus Strait.

with 53 vertical levels at the deepest point of the Black Sea. We note that this limitation may, particularly in the Bosphorus, lead to excessive vertical mixing or a widened interface thickness, which are crucial for the intrusion of the Mediterranean origin water into the Black Sea. This can be seen in their Figures 11c and 11d, for example.

In our work, we simulate the complete TSS system with a high vertical resolution unstructured grid model forced by complete
5  heat, water and momentum fluxes. A long-term six-year simulation is used to analyse the combined response of the Marmara Sea to atmospheric forcing and strait dynamics. The questions addressed in this paper are: What is the mean Marmara Sea circulation and its variability in a long-term simulation? What are the effects of the atmospheric forcing on the Marmara Sea dynamics and circulation?

The paper is organised as follows: In the next section, we document the model setup and the details of the experiment. In
10  section 3, the validation of the water mass structure and sea level differences along the TSS are demonstrated. The resulting volume transports through the straits, and the kinetic energy and circulation in the Marmara Sea, are presented. Finally, in section 4, we summarise and discuss the results.





## 2   Model Setup

Models for solving the dynamical equations for an unstructured mesh using finite element or finite volume methods have been implemented for many idealistic applications (White et al., 2008; Ford et al., 2004), and in realistic coastal ocean studies (Zhang et al., 2016; Federico et al., 2017; Stanev et al., 2017). An obvious advantage of using an unstructured mesh model

is the varying resolution, which allows for a finer mesh resolution in coastal areas than in the open ocean. The general ocean circulation model used in this study is the Finite Element Sea-ice Ocean Model (FESOM). FESOM is an unstructured mesh ocean model using finite element methods to solve hydrostatic primitive equations with the Boussinesq approximation (Danilov et al., 2004; Wang et al., 2008). We use the initial implementation of Gürses et al. (2016).

The model domain extends zonally from 22.5°E to 33°E and meridionally from 38.7°N to 43°N, covering a total surface

area of 1.52 x $10^{11}$ m$^2$ (Fig. 1). The mesh resolution is as fine as 65 m in the Bosphorus and 150 m in the Dardanelles. In the Marmara Sea, the resolution is always finer than 1.6 km and is not coarser than 5 km in the Black Sea and the Aegean Sea. The water column is discretized by 110 vertical z-levels. The vertical resolution is 1 m in the first 50 m depth and increases to 65 m at the bottom boundary layer in the deepest part of the model domain.

Model equations and parameters are documented in appendix A. The current model implementation considers closed lateral

boundaries. Therefore, volume and salinity conservations are imposed to prevent drifts in the tracer fields. Our approach for volume and salinity conservations is described in appendix B.

The experiment detailed below was conducted over six years, commencing on 1 January 2008 and continuing until 31 December 2013. The initial fields were obtained after a three-month integration of a lock-exchange case, which was initialised from different temperature and salinity profiles in the basins of the Black, Marmara and Aegean Seas (Gürses et al., 2016;

Sannino et al., 2017). Fine mesh resolution and energetic flow structures in the straits require small time steps so the CFL condition is not violated. The time step is thus set to 12 s throughout the integration. The simulation was forced by atmospheric fields provided by ECMWF with 1/8° resolution. The forcing data cover the whole experiment period with a time frequency of six hours. Precipitation data are obtained from monthly CPC Merged Analysis of Precipitation (CMAP, Xie and Arkin (1997)) and interpolated to the ECMWF grid as daily climatology.

The annual mean of wind, wind stress and wind stress curl for the simulation period are shown in Fig. 2. Wind stress, $\tau$, is calculated as

$$\tau = \rho_0 C_d \mid \mathbf{u}_{wind} \mid \mathbf{u}_{wind} \tag{1}$$

where $C_d$ is the drag coefficient and $\mathbf{u}_{wind}$ is the wind velocity. The annual mean wind fields are northeasterlies which were strongest in 2011. $\tau$ was higher than 0.04 $Nm^{-2}$ in the central-north Marmara Sea in 2009. It then expands in a north-south

direction, exceeding 0.05 $Nm^{-2}$ in the central basin, in 2011 and then weakens again in 2013. The wind stress curl is a dipole shaped by the northeasterlies and is negative in the north and west, and positive in the south and east of the Marmara Sea. In 2011, the wind stress curl in the coastal zones was more intense than in the other years.

A surface area of 2.22 x $10^{10}$ m$^2$ north of 42.5°N in the Black Sea functions as a buffer zone (the grey shaded area in Fig. 1). This zone is utilised to provide required water fluxes for a realistic barotropic flow through the Bosphorus. The model





**Figure 2.** Annual mean of wind velocity (ms$^{-1}$, arrows), wind stress ($10^{-2}$Nm$^{-2}$, black contours) and wind stress curl ($10^{-6}$Nm$^{-3}$, shades) in the Marmara Sea for each year from 2008 (top left) to 2013 (bottom right). The coastline is overlaid in blue.

is forced by a climatological runoff in the Black Sea, which is essential to generate realistic sea level differences between the compartments of the system (Peneva et al., 2001). The monthly runoff climatology for water fluxes was obtained from Kara et al. (2008) and are the same for all the six years. The surface salinity at the buffer zone was relaxed to a monthly climatology, computed from a 15-year simulation by the Copernicus Marine Environment Monitoring Service Black Sea circulation





| Month | Jan | Feb | Mar | Apr | May | Jun |
|---|---|---|---|---|---|---|
| $R(km^3/yr)$ | 260.3 | 281.7 | 333.9 | 404.1 | 417.6 | 353.6 |
| $S^*(psu)$ | 18.97 | 18.96 | 18.91 | 18.88 | 18.74 | 18.74 |
| Month | Jul | Aug | Sep | Oct | Nov | Dec |
| $R(km^3/yr)$ | 292.5 | 231.2 | 198.7 | 196.2 | 223.0 | 254.2 |
| $S^*(psu)$ | 18.87 | 18.98 | 18.90 | 18.92 | 18.94 | 19.02 |

**Table 1.** Monthly Black Sea river discharges and salinity relaxation values.

model (Storto et al., 2016). The salinity relaxation time is approximately two days. Although this is a strong constraint, it is required to prevent the surface salinity from decreasing in the buffer zone due to the excessive amount of fresh water input. The climatological values used for runoff and salinity relaxation are shown in Table 1.

## 3   Results

In this section, we present the results obtained from a simulation of the TSS between 2008 and 2013. The focus is on the Marmara Sea and the straits, but we also consider the adjacent basins when necessary. We provide details of the main water mass characteristics of the system and our validation against the observations. The sea level differences and the volume transports through the Bosphorus and Dardanelles are analysed. Although we will include results demonstrating the response of the system to daily atmospheric events, we focus on the interannual changes in the TSS. Therefore, we consider only annual

means in time averages. The computed time averages for the simulation are for the period 2009-2013, as the first months of 2008 are considered as an initial spin-up period.

### 3.1   Surface Heat and Water Fluxes

The monthly averages of water fluxes and net heat flux in the Marmara Sea are shown in Fig. 3. The runoff is relatively small and is thus approximated to be zero in the Marmara Sea. Evaporation fluctuates between $-5.1 \times 10^{-8}$ and $-5 \times 10^{-9}$ m/s with an

absolute minimum in March and a maximum in July. Minimum precipitation is $1 \times 10^{-9}$ m/s in July while the maximum is $3.3 \times 10^{-8}$ m/s in December. The resulting net water flux $P$ - $E$ varies between $-4.7 \times 10^{-8}$ and $2.5 \times 10^{-8}$ m/s. The net heat flux in the Marmara Sea is calculated in the range of $-123.3$ W/m$^2$ to $138.4$ W/m$^2$ with minimums and maximums in December-January and May-June, respectively.

The daily buoyancy fluxes were averaged between 2009-2013 and are shown in Fig. 4. The buoyancy flux was computed

using the formula:

$$Q_b = \frac{g\alpha}{\rho_0 C_w} Q_H - \beta S_0 g(E - P - R) \tag{2}$$



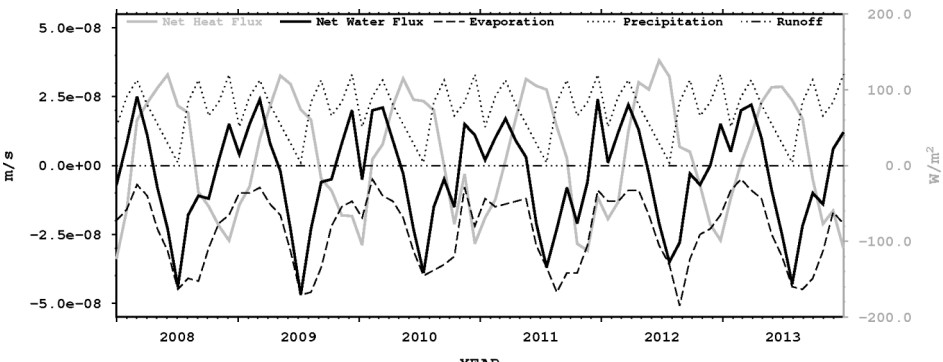

**Figure 3.** Monthly averaged net heat (grey) and water (black) fluxes in the Marmara Sea, along with evaporation (dashed) and precipitation (dotted). Runoff is zero in the Marmara Sea. The grey vertical axis (right) is for heat flux and the black vertical axis (left) for water fluxes.

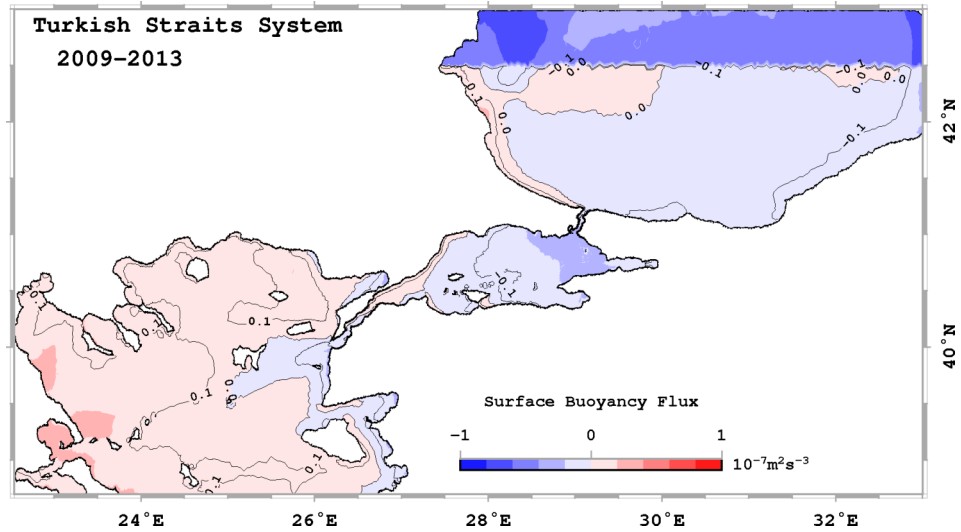

**Figure 4.** Mean of surface buoyancy fluxes for 2009-2013. A negative value is the buoyancy flux down into the ocean.

where $\alpha$ and $\beta$ are thermal and haline expansion coefficients, $Q_H$ is the heat flux, $C_w$ is the specific heat capacity, $S_0$ is the surface salinity and $E - P - R$ is the water flux.

The Black Sea and the Marmara Sea gain buoyancy except for a small area near their western coasts (Fig. 4), while the Aegean Sea has a buoyancy loss except near the Dardanelles exit and the Anatolian coast. The average buoyancy flux changes

5  between -7x10$^{-8}$ and 3.4x10$^{-8}$ m$^2$s$^{-3}$ over the domain. It does not show significant spatial differences interannually, but the gradient between the Aegean and Black Seas was stronger in 2011 than in the other years (not shown).





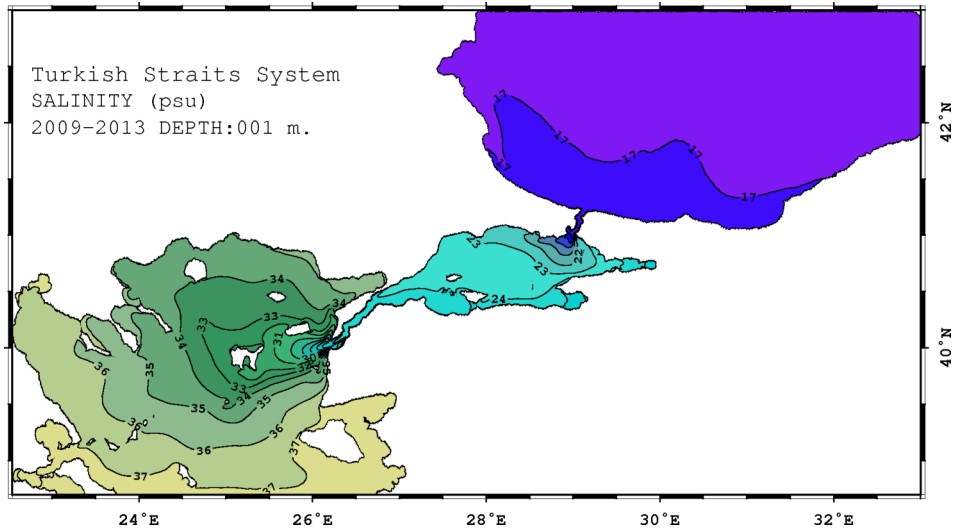

**Figure 5.** The mean sea surface salinity for 2009-2013. Contours are overlaid with 1 psu interval.

### 3.2 Water Mass Structure and Validation

The surface salinity ranges from 16 to 38 psu over the whole domain (Fig. 5). The surface waters leave the Bosphorus and the Dardanelles with salinities of about 21 psu and 27 psu, respectively. The surface salinity in the northern Marmara Sea is less than 23 psu and increases to 25 psu in the south. Long-term measurements from 1986 to 1992 in the Marmara Sea (Beşiktepe

5   et al., 1994) suggest a salinity ranging between 23±2 psu at the surface, which is satisfied by the simulation.

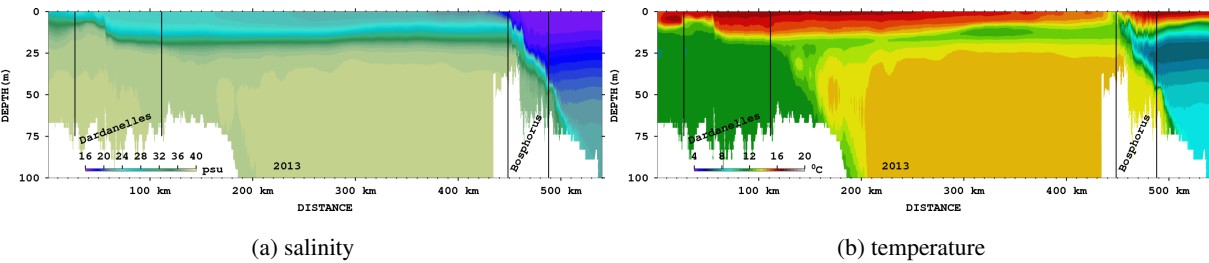

(a) salinity          (b) temperature

**Figure 6.** Annual mean of a) salinity and b) temperature for 2013 along the thalweg.

The vertical structure of the time mean salinity and temperature along the thalweg (see Fig.1) is shown for the last year of the integration in Fig. 6. The depth of the interface between the upper and lower layers is stationary throughout the simulation and is located at a depth of around 20 m. The water column salinity is mixed below 25 m in the Marmara Sea whereas it is stratified in the straits. Altıok et al. (2012) reported a cold tongue in the Bosphorus in June-July between 1996-2000, with a

10  temperature of about 11-12°C, extending to the Marmara Sea. This cold tongue is reproduced in the simulation (Fig. 6b) and emerges as a cold intermediate layer (CIL) at the position of the halocline.





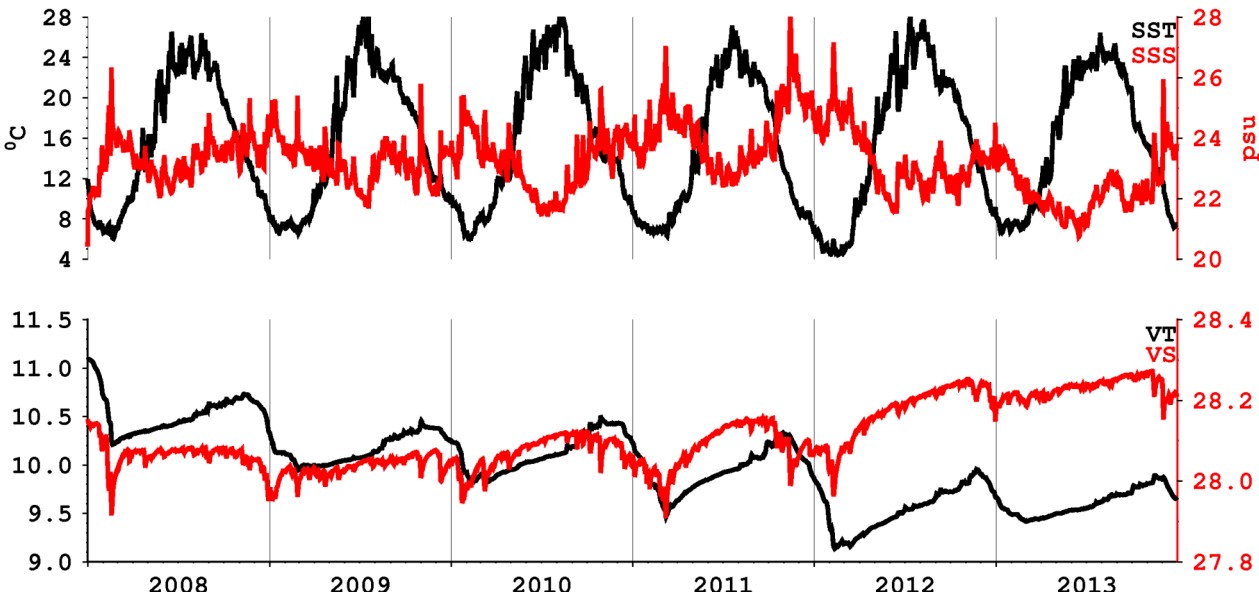

**Figure 7.** Daily time series of mean sea surface temperature (SST), mean sea surface salinity (SSS), volume mean temperature (VT), and volume mean salinity (VS) in the Marmara Sea. Temperature and salinity are shown in black and red, respectively.

The mean sea surface temperature in the Marmara Sea fluctuates between 4.4 - 28.3°C (Fig. 7). Surface mean salinities are lower in the spring and early summer than at other times of year. The volume mean temperature decreases from 11°C to 9-10°C and varies seasonally.

Three datasets of in-situ CTD observations were collected by R/V Bilim2 from the Institute of Marine Sciences (IMS/METU[1]), in 4-11 April 2008, 1-4 October 2008 and 18-23 June 2013 are used to validate the simulation.

| | April 2008 | | October 2008 | | June 2013 | |
|---|---|---|---|---|---|---|
| psu / °C | Salinity | Temperature | Salinity | Temperature | Salinity | Temperature |
| | 1.34 | 0.77 | 1.71 | 1.09 | 3.18 | 2.09 |

**Table 2.** Mean RMS of salinity and temperature error with respect to CTD measurements in April and October 2008 and June 2013.

Figure 8 shows the spatial distribution of the salinity and temperature RMS errors in the first 50 m of the water column in October 2008. The error is higher in the eastern Marmara Sea close to the Bosphorus. The mean RMS errors of temperature and salinity are listed in Table 2. The errors are similar in April and October 2008, but notably increase after six years of

---

[1] Two are from European SESAME-Southern European Seas: Assessing and Modeling Ecosystem Changes Integrated Project/ FP6. The other dataset is from the subsequent PERSEUS: Policy-oriented marine Environmental Research for the Southern European Seas, funded by the EU under the FP7 Theme 'Oceans of Tomorrow' OCEAN.2011-3 Grant Agreement No. 287600 project.





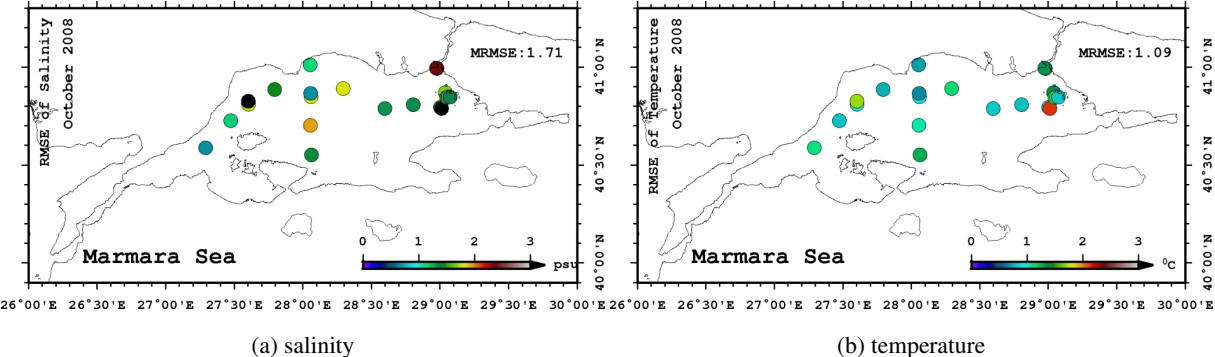

|  |  |
|:-:|:-:|
| (a) salinity | (b) temperature |

**Figure 8.** Spatial distibution of RMS of a) salinity and b) temperature error computed at the top 50 m. of each CTD cast in October 2008

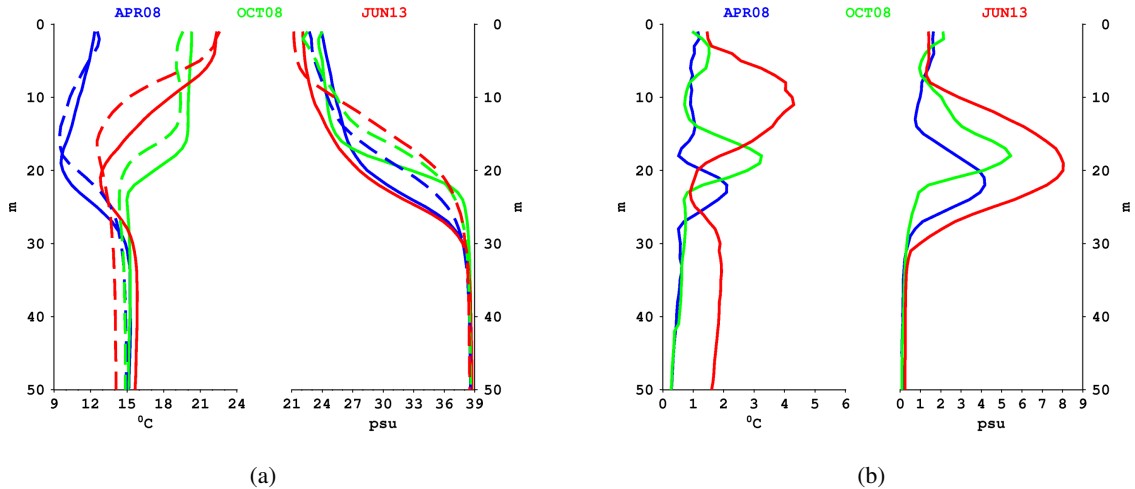

|  |  |
|:-:|:-:|
| (a) | (b) |

**Figure 9.** a) Mean temperature and salinity profiles of observations (dashed) and simulation (line) and b) vertical distributions of RMS errors in April 2008 (blue), October 2008 (green) and June 2013 (red).

integration in June 2013, which can be expected after such a long integration and, spatially, the errors are this time higher in the Dardanelles side of the Marmara Sea than the Bosphorus side. The RMS error is larger than those found in operational models of the Mediterranean Sea (Oddo et al., 2009), but this is due to a thermocline and halocline vertical shift, as shown in Fig. 9a. The vertical mixing and the missing interannual variability of the Black Sea runoff probably account for this. The model performs substantially better at the surface and below 30 m in depth than it does at the depth of the interface between the upper and lower layers of around 20 m (Fig. 9b). Temperature RMS errors are highest around the seasonal thermocline in June 2013, while at the same level as the halocline in the other two months.





### 3.3 Sea level and volume fluxes through the straits

The time mean of the sea surface height across the system is shown in Fig. 10 for the 2009-2013 period. The SSH is approximately 0.12 m in the Black Sea and -0.12 m in the Aegean Sea. In the Marmara Sea, SSH is higher in the western basin. The differences between the two ends of the Bosphorus are approximately 0.18 m and in the Dardanelles are about 0.11 m.

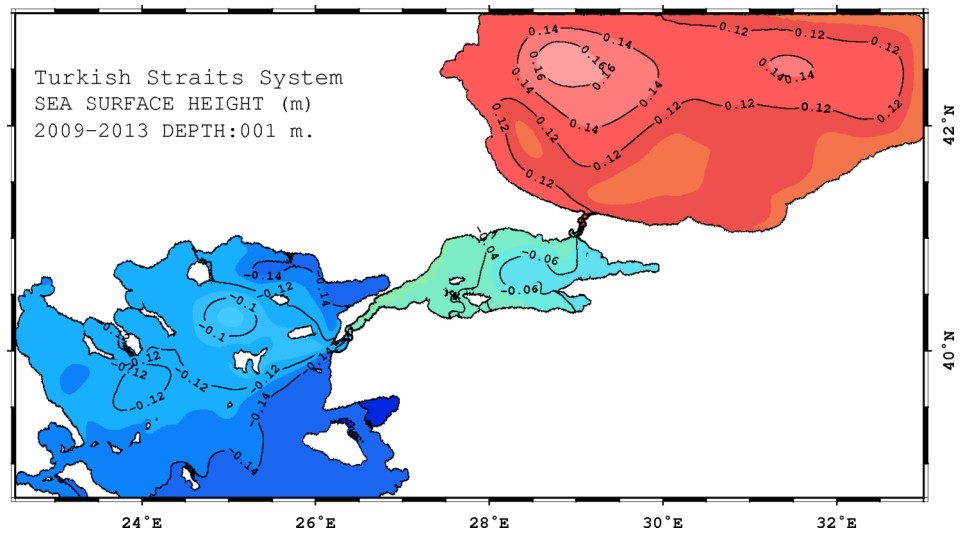

**Figure 10.** The mean sea surface height for 2009-2013. Contours are overlaid with 2 cm interval.

Moller (1928) measured the sea level differences between the two ends of the Dardanelles and the Bosphorus as 7 and 6 cm, respectively. Gunnerson and Ozturgut (1974) and Büyükay (1989) found the sea level difference in the Bosphorus to be 35 cm for the 1966-1967 period and 28-29 cm for 1985-1986 period, respectively. Bogdanova (1969) gave a sea level difference of 42 cm between the northern Black Sea (Ialta) and southern coast of Turkey (Antalya) with high seasonal variability. Alpar et al. (2000) suggested a mean sea level difference of 55 cm between the Black Sea entrance of the Bosphorus and the Aegean

entrance of the Dardanelles for the 1993-1994 period. We cannot assess the accuracy of our model against these values in the literature with such a wide range of variability.

As shown in Fig. 11, we compare the sea level difference between Yalova (Marmara Sea, see Fig. 1) and Şile (Black Sea) with the time series of tide gauge measurements collected between 2008 and 2011 (Tutsak et al., 2016). The response of the model is weaker during the arrival phase of severe storms, corresponding to southwesterly winds known as 'Lodos', when

the sea level difference becomes negative in the observations; i.e., the sea level in the Marmara Sea is higher than in the southwestern Black Sea. The higher sea level in the Marmara Sea often results in short term blocking and reversal of the Bosphorus upper layer flow (called Orkoz). One such event was studied by Book et al. (2014) during the strong atmospheric cyclone passage of Nov 22, 2008 (Fig. 11). The signal of this event is captured in the sea level difference measurements (blue) and successfully reproduced by the model (red), though with a smaller amplitude. However, the event of 21 August 2010 is



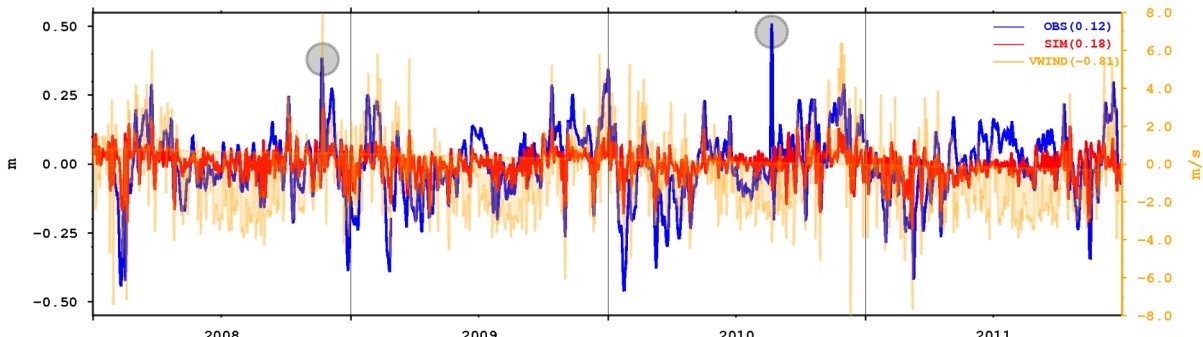

**Figure 11.** Sea level difference (SLD) between Yalova and Şile and daily averages of the meridional wind speed at B1 (See Fig. 1) between 2008-2011. The SLDs in the in-situ observation (OBS) and the simulation (SIM) in meters are shown by the blue and red lines, respectively. The meridional wind speed is in orange in meters per second. The four-year means, which are subtracted from the time series, are shown in the legend. A positive SLD means the sea level is higher in Yalova than in Şile. A positive wind speed means the wind direction is northward. The grey circles mark the Orkoz events in Nov. 22, 2008 and Aug. 21, 2010 as suggested by the observations.

absent in the simulation as the atmospheric forcing (orange) shows no signal of a severe southerly storm. The accuracy of the atmospheric forcing is in this case limiting the correct oceanic response.

The salinity structure on Nov 15, 2008 represented in Fig. 12 along the thalweg line of Fig. 1 corresponds to the situation typically observed in the Bosphorus with a normal range of net flow. The features of the salinity distribution shown in Fig. 12a

are similar to those shown by the measurements of Özsoy et al. (2001) and Gregg and Özsoy (2002), and computed by Sözer and Özsoy (2017a) and Sannino et al. (2017) respectively, in stand-alone Bosphorus and integrated TSS models of exchange flows under a medium range of net flows excluding the effects of atmospheric forcing. The interface becomes thinner in the buoyant Bosphorus Jet flowing into the Marmara Sea and at the northern sill, where the lower layer reaches supercritical speeds through the hydraulic control, followed by a series of hydraulic jumps (Dorrell et al., 2016).

In comparison, the next section in Fig. 12b reflects the situation predicted in the Bosphorus on Nov 22, 2008. Under the conditions of an extreme Orkoz event, the upper layer flow of the Bosphorus becomes completely blocked. In rare instances the entire Bosphorus has been observed to flow towards the Black Sea, as reported after Nov 22, 2008, which was covered by a period of extensive measurements. ADCP measurements in the middle of the Bosphorus indicated a flow towards the Black Sea over the entire depth of the Strait, with superposed minor oscillations (Tutsak et al., 2016) during the event (Jarosz et al.,

2011a; Book et al., 2014). The increased flow of dense water of mixed Mediterranean and TSS origin was found to cascade over the Black Sea shelf and propagate over large distances across the interior of the Sea at intermediate depths (Falina et al., 2017). The model results in Fig. 12 show increased vertical mixing in the upper layer up in the middle of the Bosphorus, but not all the way to the northern end as indicated by the measurements. The upper layer has been pushed north of the southern contraction, suggesting that the hydraulic control there has been lost. However, the thin interface layer on top of the northern

sill and the flow north of it continues to preserve its shape, suggesting continued hydraulic control at the northern sill.

All the above features are consistent with the findings obtained from idealised and realistic models of the Bosphorus provided in Sözer and Özsoy (2017a) and the TSS model experiments of Sannino et al. (2017).





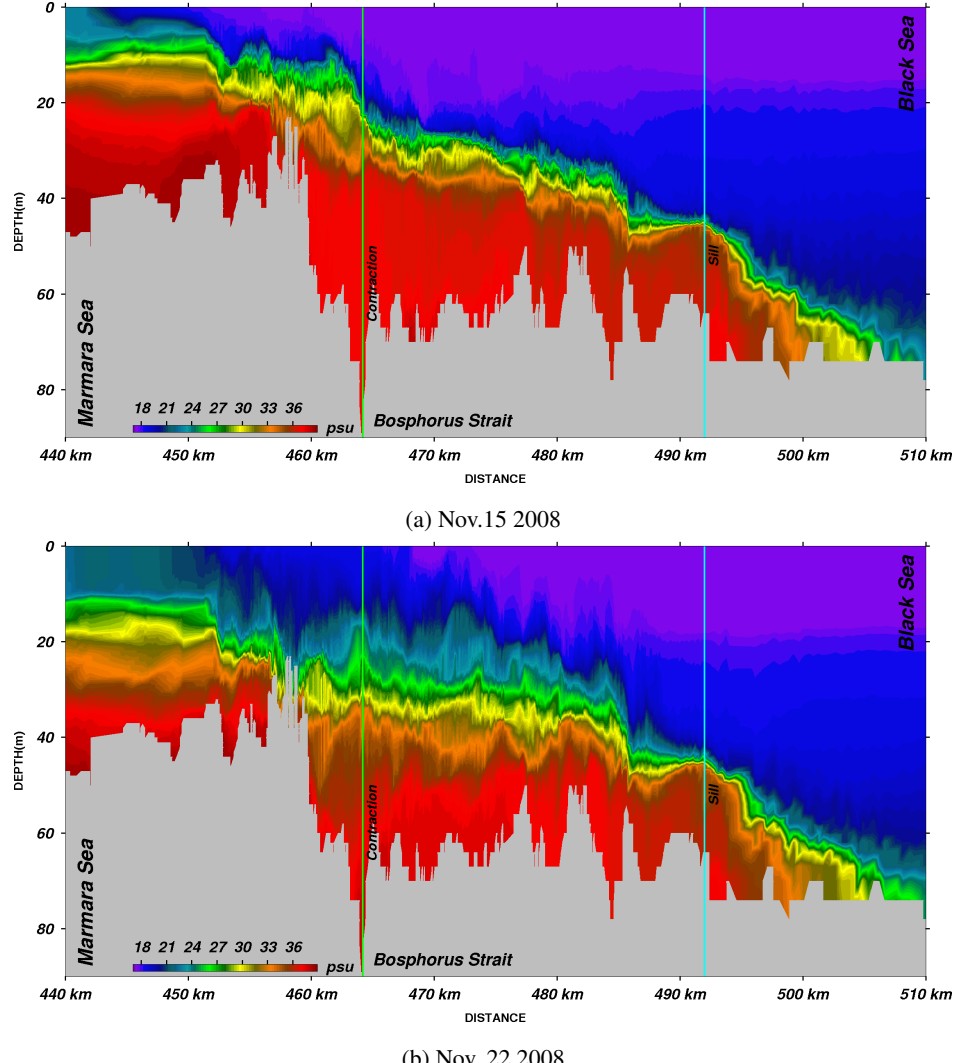

(a) Nov.15 2008

(b) Nov. 22 2008

**Figure 12.** Cross-section of salinity in the Bosphorus Strait on a) Nov 15 b) Nov 22 2008 along the thalweg between 440-510 km. Contraction and the northern sill locations are marked with green and cyan lines, respectively, as shown in Fig. 1.

Upper layer velocity in the southern exits of both the Bosphorus and Dardanelles are generally higher than their northern exits, due to the hydraulic controls exerted at the constrictions in the middle of both straits and the expansion area in the mouth (Sözer and Özsoy, 2017a). Conversely, the lower layer velocity has much higher maxima at the northern exits in both straits, and are roughly 0.2 m/s less than the measurements by Jarosz et al. (2011a, 2012). The upper layer maximum velocities are in accordance with most observations. The depth of the zero velocity level varies significantly in the exits of each strait and are listed in Table 3. These depths are consistent with the recent approximations reported in Jarosz et al. (2011a, 2012) for the



| | UL $|\mathbf{u}_{max}|$ $(m/s)$ | LL $|\mathbf{u}_{max}|$ $(m/s)$ | Interface depth (m) |
|---|---|---|---|
| Northern Bosphorus | -0.35 | 1.4 | 40 |
| Southern Bosphorus | -1.85 | 0.63 | 10 |
| Northern Dardanelles | -1.0 | 0.78 | 20 |
| Southern Dardanelles | -1.8 | 0.5 | 10 |

**Table 3.** Along-strait maximum velocity for the upper layer (UL) and lower layer (LL) and interface depth in each strait exit. Units are in $m/s$ and $m$ for the velocity and depth, respectively. A negative value means the flow in the direction from the Black Sea to the Aegean Sea.

northern Bosphorus, the southern Bosphorus, the northern Dardanelles and the southern Dardanelles, which are 39 m, 13.5 m, 22 m and 13 m, respectively.

| $km^3/yr$ | Net | UL | LL |
|---|---|---|---|
| Northern Bosphorus | -147.7 | -283.3 | 135.5 |
| Southern Bosphorus | -148.5 | -307.7 | 159.2 |
| Northern Dardanelles | -104.2 | -343.8 | 239.6 |
| Southern Dardanelles | -102.3 | -443.6 | 341.3 |

**Table 4.** Annual mean of net, upper layer and lower layer volume fluxes ($km^3/yr$) for the whole simulation period. A negative value means the flux is in the Black Sea-Aegean Sea direction.

The net, annual mean upper layer and lower layer volume fluxes are given in Table 4 and the daily and monthly averages shown in Figure 13. The estimations from Jarosz et al. (2011b, 2013) from direct measurements are also indicated. The mean net volume fluxes through the straits compare well with the observations between 2 September 2008 and 5 February 2009 for the Bosphorus, and 1 September 2008 to 31 August 2009 for the southern Dardanelles. However, the variability in time is not as high as in the observations (Figure 13). In the northern Dardanelles, there is a large discrepancy between the simulated net fluxes and the estimates from observations.

Sannino et al. (2017) and Özsoy and Altıok (2016) both have similar disagreements with the measurements of Jarosz et al. (2013) and have concluded that the discrepancies could be a result of measurement or computational inaccuracies at the wide northern section of the Dardanelles, where the instrument data were located.

The historical estimates of the Dardanelles layer volume fluxes show much higher values than our simulation (Table 5). The changes found in the layer flux from one end of the strait to the other have been attributed to turbulent entrainment processes, which transport water and properties across the hypothesised layer interface (Ünlüata et al., 1990; Özsoy et al., 2001; Özsoy and Altıok, 2016). The larger disagreement of baroclinic volume fluxes compared to the net fluxes, and the lower estimations of the model layer fluxes, suggest that bottom friction parametrization is too strong and possibly a problem in the vertical mixing submodel chosen.







(a) Northern Bosphorus

(b) Southern Bosphorus

(c) Northern Dardanelles

(d) Southern Dardanelles

**Figure 13.** Daily upper layer (blue, UL), lower layer (red, LL) and net (grey, NET) volume fluxes through northern Bosphorus (NB), southern Bosphorus (SB), northern Dardanelles (ND) and southern Dardanelles (SD) in km³yr⁻¹. Monthly and six-year averages are overlaid with a darker tone of the same colour. The monthly averages of volume fluxes computed by Jarosz et al. (2011b, 2013) are shown in green for the period of observations.



|  | Northern Dardanelles | | | Southern Dardanelles | | |
| --- | --- | --- | --- | --- | --- | --- |
| Annual Fluxes ($km^3/yr$) | UL | LL | Net | UL | LL | Net |
| Ünlüata et al. (1990) | -865.9 | 566.0 | -299.9 | -1257.0 | 966.5 | -290.5 |
| Beşiktepe et al. (1994) | -846.7 | 546.8 | -299.9 | -1217.6 | 917.7 | -299.9 |
| Özsoy and Ünlüata (1997) | -829.7 | 529.8 | -299.9 | -1179.7 | 879.8 | -299.9 |
| Tuğrul et al. (2002) | -918.6 | 597.9 | -320.7 | -1330.5 | 1009.7 | -320.8 |
| Kanarska and Maderich (2008) | -666.7 | 391.0 | -275.7 | -1224.2 | 946.0 | -278.2 |

**Table 5.** Annual means of volume fluxes ($km^3/yr$) through the Dardanelles estimated by different studies. UL, LL and Net stand for upper layer, lower layer and net fluxes, respectively. A negative value means the volume flux is from the Marmara Sea to the Aegean Sea.

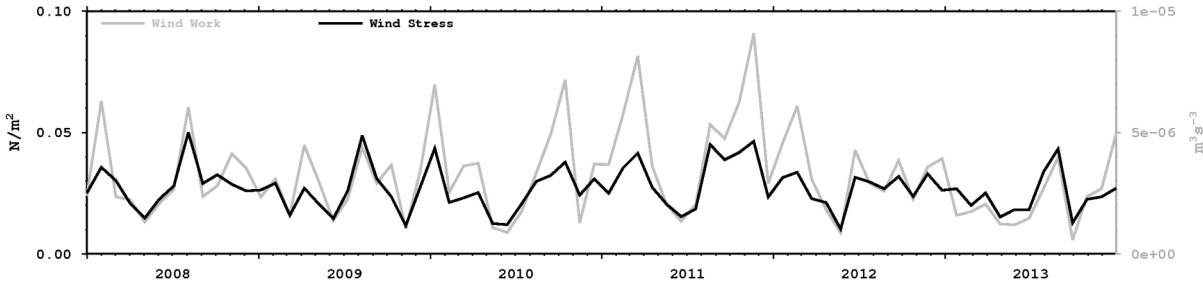

**Figure 14.** Monthly time series of the wind work ($m^3 s^{-3}$) and wind stress ($N m^{-2}$) in the Marmara Sea. The wind work normalised by the surface area of the Marmara Sea is shown in grey in the right vertical axis. The wind stress is the black curve and its values are shown on the left vertical axis.

### 3.4 Marmara sea dynamics and circulation

Using the six-year simulation, it is now possible to estimate the kinetic energy input by the wind in the Marmara Sea.

The time series of monthly mean wind stress is shown in Fig. 14. It exhibits interannual differences with a mean of about 0.03 $N/m^{-2}$ and a maximum of around 0.05 $Nm^{-2}$ in August 2008. The monthly mean is highly variable and there is not a

5   well-defined seasonal cycle between 2008-2013. The wind work normalised by the surface area is computed as:

$$\frac{1}{\rho_0 A} \int_A \tau \cdot \mathbf{u}_s \, dx dy \tag{3}$$

where $\rho_0$ is the surface density, $\tau$ is the wind stress, $\mathbf{u}_s$ is the current velocity at the surface and $A$ is the surface area of the Marmara Sea. The wind work is positively correlated with the wind stress. It is highest in 2011 and the maximum monthly mean wind work is $9.1 \times 10^{-6}$ m$^3$s$^{-3}$ in the autumn of 2011.

10   To compare with the other marginal seas described in Cessi et al. (2014), we normalise the right hand side of the equation 3 by the volume of the Marmara Sea instead of the surface area. The six-year mean of the wind work is computed as $1.09 \times 10^{-8}$ m$^2$s$^{-3}$, one order of magnitude higher than the Mediterranean Sea. The wind work in the Baltic Sea was computed to be $9.15 \times 10^{-9}$ m$^2$s$^{-3}$, which is comparable to the Marmara Sea but is still lower.





The resulting volume-mean kinetic energy in the Marmara Sea normalised by the unit mass is calculated as 0.006 $m^2 s^{-2}$ for six years. The daily mean kinetic energy time series reveals that severe atmospheric events are able to energise the basin up to 0.03 $m^2 s^{-2}$ (not shown). The monthly volume-mean kinetic energy averages fluctuate between 0.005-0.01 $m^2 s^{-2}$. It is higher in winter and early spring, whereas it is always below the mean in summer. The highest kinetic energy inputs are in October 2010, April 2011 and November 2011. The kinetic energy is higher in the upper layer of the water column. The time-mean of surface kinetic energy is about 0.03 $m^2 s^{-2}$. Daily surface averages are capable of reaching 0.2 $m^2 s^{-2}$. The monthly mean increases to approximately 0.05 $m^2 s^{-2}$ in November 2011.

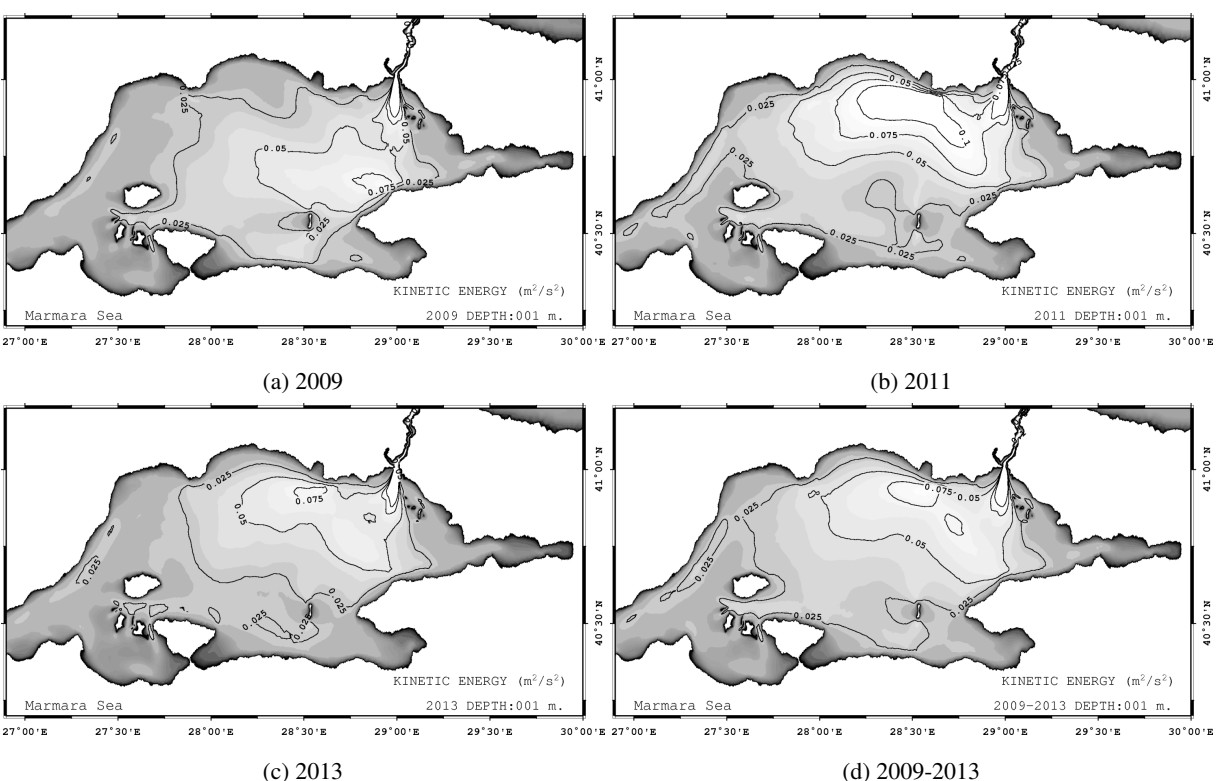

**Figure 15.** Annual mean of surface kinetic energy in the Marmara Sea for a) 2009 b) 2011 c) 2013. d) The time-mean of the 2009-2013 period. Units are in m²s⁻² as kinetic energy is normalised by the unit mass.

In the Marmara Sea, the buoyancy gain is mainly due to the Bosphorus inflow, and thus the latter competes with the wind work to change the kinetic energy of the basin, as explained by Cessi et al. (2014). However, at the surface the kinetic energy shows that the Bosphorus surface jet energises the northeastern basin in addition to the wind (Fig. 15). The kinetic energy of the Bosphorus inflow is always greater than 0.075 $m^2 s^{-2}$. In 2009, a kinetic energy maximum appears in the north of the Bozburun peninsula where the Bosphorus jet arrives. In the western basin, the kinetic energy is generally less than 0.025 $m^2 s^{-2}$. In 2011, the kinetic energy intensifies in the central north and exceeds 0.1 $m^2 s^{-2}$. Almost all of the basin, except near





the coastal areas, has a kinetic energy higher than 0.025 $m^2 s^{-2}$. Energy is mostly confined to the north-east of the central basin in 2013. The time-mean for 2009-2013 reflects the characteristics of 2011 but with less amplitude.

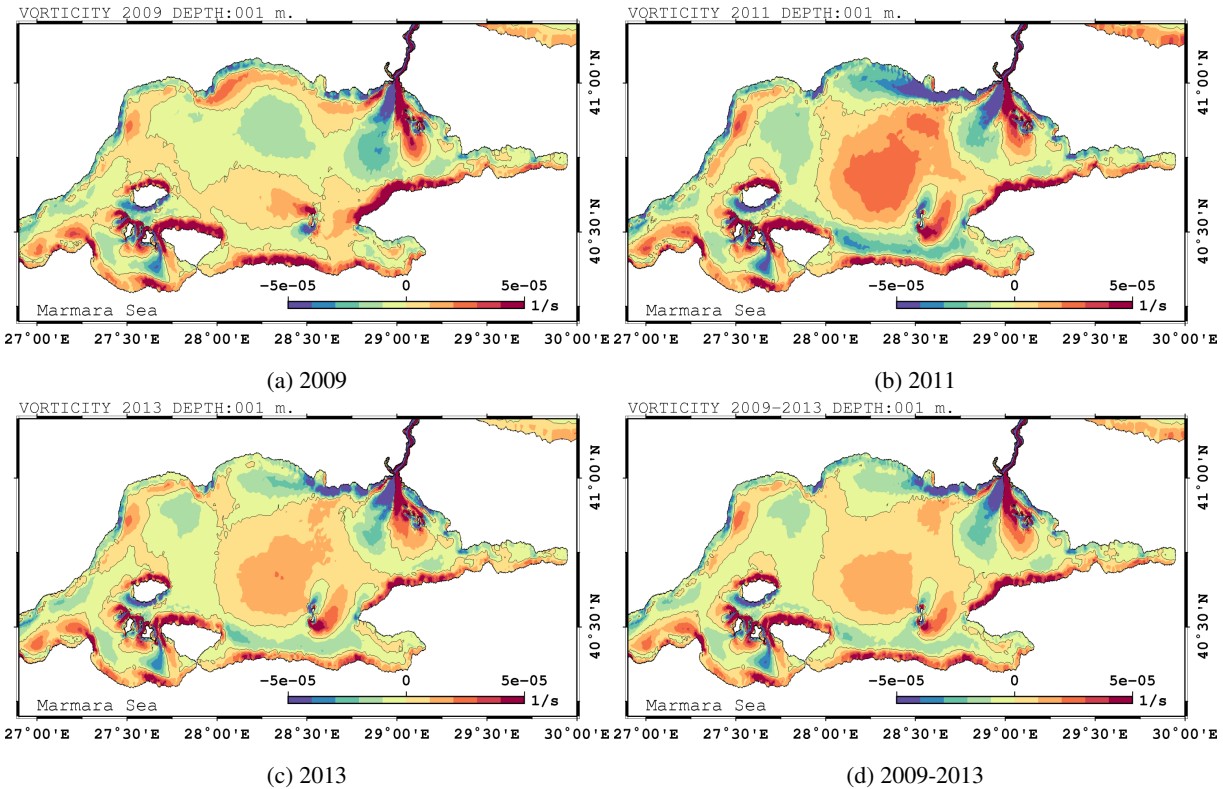

**Figure 16.** Annual mean of surface vorticity in the Marmara Sea at the surface. a) 2009 b) 2011 c) 2013, d) the surface vorticity mean for the 2009-2013 period.

The energetic Bosphorus jet generates a dipole vorticity field, which is anti-cyclonic in the west and cyclonic in the east (Fig. 16). Northern and western coasts are dominated by anticyclonic vorticity. Conversely, cyclonic vorticity dominates the southern coast. Positive and negative vorticity in the northern and southern coasts of the islands and peninsulas, respectively, are other common structures for all the years. The mean vorticity fields are consistent with upwelling favourable conditions in the southern coasts of the Marmara Sea. Primarily, a cyclonic gyre located at 28°20,N - 40°35,E forms in the central basin after 2011, showing that the circulation changed of sign?????? between 2008 and 2011.

Fig. 17 shows the annual mean of the current velocity at the surface and 30 m for 2009, 2011 and 2013, and the mean for the 2009-2013 period. The annual means of the surface circulation show two different circulation structures as was already evident from the vorticity structures. In 2009 (Fig. 17a), the Bosphorus plume reaches the Bozburun peninsula and turns west towards the middle of the basin. One branch of the flow heads north and forms an anti-cyclone close to the Trachian coast. The southern branch instead splits into two when it reaches the Marmara Island. The southwestward flow traverses the Marmara Sea after turning south, merging with the flow circulating around the islands in the southwestern Marmara Sea and eventually







(a) 2009 - surface

(b) 2009 - 30 m

(c) 2011 - surface

(d) 2011 - 30 m

(e) 2013 - surface

(f) 2013 - 30 m

(g) 2009-2013 - surface

(h) 2009-2013 - 30 m

**Figure 17.** Annual mean of current velocity in the Marmara Sea. a) 2009 at the surface, b) 2009 at 30 m, c) 2011 at the surface, d) 2011 at 30 m, e) 2013 at the surface, f) 2013 at 30 m. The means for the 2009-2013 period are shown in g) at the surface and h) at 30 m




exiting from the Dardanelles. This circulation pattern in the western Marmara Sea is persistent throughout the simulation but with different intensities. This type of circulation structure has been reported in other studies (Chiggiato et al., 2012; Beşiktepe et al., 1994). In 2011 (Fig. 17c), and the circulation in the middle of the Marmara Sea evolves into a single cyclonic structure. The shift in the circulation can be explained by the shift of the wind stress maximum towards the north (Fig. 2). Sannino et al.

(2017) demonstrates a similar cyclonic pattern in the central Marmara Sea due to the potential vorticity input by the Bosphorus. However, in our case, the main driver of the cyclone should be the wind as the volume transport through the Bosphorus is much lower than that of Sannino et al. (2017) case. Thus, in certain conditions both the wind and the Bosphorus can induce a cyclonic circulation in the Marmara Sea. The cyclonic surface circulation dominates the mean between 2009-2013. The mean surface circulation in the Sea can be sketched as in Fig. 18.

Below the pycnocline at 30 m, two main structures can be identified. An anti-cyclonic formation appears in the central basin intensifying in 2013. On the Dardanelles side, a flow enters the Marmara Sea and partially heads south east. Another structure recirculates after reaching the Marmara Island and joins the southwestward flow exiting the Marmara Sea. In 2011, a meander heading west is formed in the northern basin. This feature is not present in other years and results from the deepening of the upper layer (not shown) due to stronger wind stress.

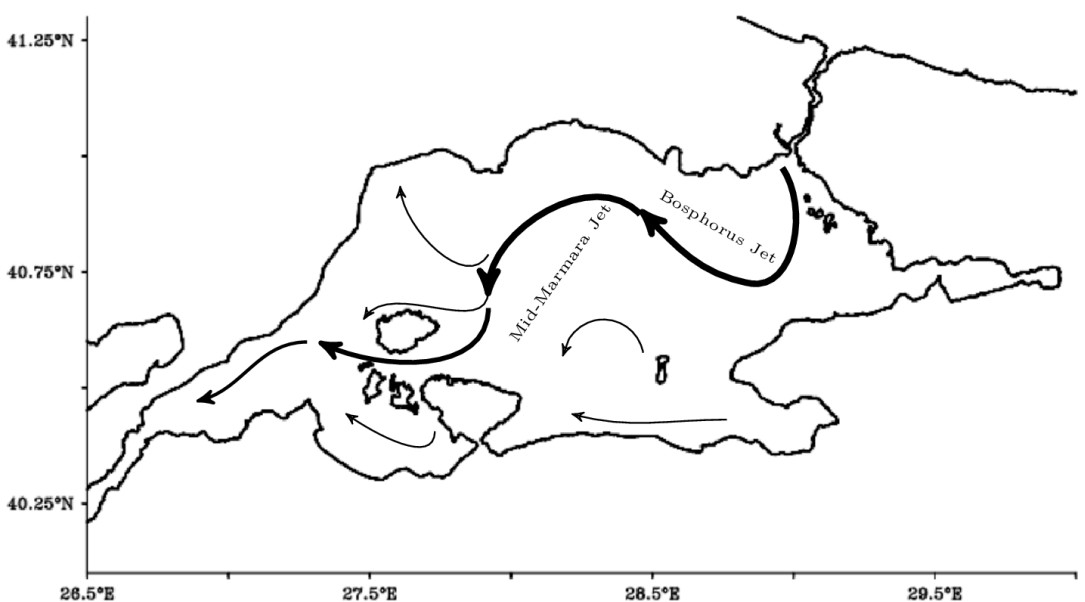

**Figure 18.** Schematic representation of the surface mean circulation in the Marmara Sea for the 2009-2013 period. The thickness of curves shows the relative intensity of the associated current.

The mean circulation of the Marmara Sea, schematically shown in Fig 18, is dominated by the Bosphorus jet and the mid-Marmara jet meandering cyclonically. The mid-Marmara jet is divided into three before reaching the Marmara Island, and one branch heads to the north west and other two reach the Dardanelles after circulating around the island. Finally, weak currents move in an east-west direction in the southern coast of the basin.





## 4 Summary and discussion

We present a six-year simulation of the Turkish Straits System, with a highly resolved unstructured mesh model of the TSS using realistic atmospheric forcing to identify the circulation structure and the interplay between the atmospheric forcing and the Bosphorus input. The results demonstrate that a realistic representation of the pycnocline from the two Straits to the

Marmara Sea is possible. The model is capable of reproducing the historically reported water mass structure of the Marmara Sea. Model errors peak at the halocline and thermocline depths, where small changes in the interface depth induce greater error.

    The Strait volume transports have been compared with the observations. The net volume transports in the Bosphorus agree with the estimates based on the observations. The model solution departs from the observations in the northern Dardanelles,

but closer agreement is found elsewhere. The baroclinic transports show larger discrepancies between the observations and the model, possibly because of uncertainties in the bottom boundary layer dissipation mechanisms and turbulence parameterization. The model is capable of simulating blocking events in the straits during severe storm passages, to the extent that such storms are present in the atmospheric dataset, as shown for the 22 November 2008 case.

    The circulation in the Marmara Sea shows two patterns in the interannual time scales. The first is dominated by the buoyant

plume of the Bosphorus to the south around an anticyclonic circulation structure in the eastern and northern parts of the basin. This type of circulation has been observed and modelled by several earlier studies (Beşiktepe et al., 1994; Chiggiato et al., 2012; Sannino et al., 2017). The other circulation structure includes a cyclone in the central basin because of intensifying and expanding wind stress over the Marmara Sea. Small scale vortices are also formed in various parts of the basin and a larger one appears in the northwest after 2011. The cyclonic gyre in the central Marmara Sea is shown numerically only by Sannino

et al. (2017), in the case of extreme net volume flux through the Bosphorus. However here we show that the wind can produce the cyclonic circulation in addition to the Bosphorus jet.

    The long-term simulation with atmospheric forcing made it possible to evaluate the wind energy input and compute the kinetic energy in the Marmara Sea. The wind work in the Marmara Sea is shown to be even higher than the Baltic Sea. The high energy input from the wind significantly increases the kinetic energy in the Marmara Sea. During severe storms, kinetic

energy can increase by 10 times the time-averaged value. The annual mean of kinetic energy in the regions under the influence of the wind forcing can also exceed that from the Bosphorus jet, depending on the wind stress structure.

    In our modelling approach, we focused our attention on the TSS proper, which is the central domain including the Straits and the Marmara Sea, while assigning a limited storage role to the truncated exterior domains of the Black and Aegean Seas. This represents a trade-off between a truthful reproduction of the TSS circulation and the ability to impose far field boundary

conditions in artificial closed basins at the two ends, using a similar strategy to that of Sannino et al. (2017). In future studies, we expect to obtain improved results by incorporating lateral open ocean boundary conditions in the Aegean and Black Seas. The skill of the model predictions also appeared sensitive to the accuracy of the atmospheric forcing used in the simulation. Within the relatively small domain of the TSS, an improved representation of the atmospheric forcing, particularly during the severe storms frequenting the region in winter, appears to be essential for improving its skill.



Overall, the results suggest further directions for long-term modelling in the TSS. We have demonstrated that wind forcing determines the surface circulation together with the Bosphorus inflow in the Marmara Sea. A higher resolution atmospheric forcing and better representation of the Black Sea water budget, by using open lateral boundaries, would improve the model solution particularly for the volume flux and the salinity flux estimations.

## 5 Appendix A: Model Equations

FESOM solves the standard set of hydrostatic primitive equations with the Boussinesq approximation (Wang et al., 2008).

The momentum equations are:

$$\partial_t \boldsymbol{u} + \boldsymbol{v} \cdot \nabla_3 \boldsymbol{u} + f\hat{k} \times \boldsymbol{u} = -\frac{1}{\rho_0}\nabla p - g\nabla\eta - \nabla A_h \nabla(\nabla^2 \boldsymbol{u}) + \partial_z A_v \partial_z \boldsymbol{u} \tag{A1}$$

where $\boldsymbol{u} = (u, v)$ and $\boldsymbol{v} = (u, v, w)$ are 2D and 3D velocities, respectively in the spherical coordinate system, $\rho_0$ is the mean density, $p$ is the hydrostatic pressure obtained through integrating the hydrostatic relation (A3) from z = 0, $g$ is the gravitational acceleration, $\eta$ is the sea surface elevation, $f$ is the Coriolis parameter and $\hat{k}$ is the vertical unit vector. $\nabla$ and $\nabla_3$ stand for 2D and 3D gradients or divergence operators, respectively. The horizontal and vertical viscosities are denoted by $A_h$ and $A_v$.

The Laplacian viscosity is known to be generally too damping and strongly reduces the eddy variances of all fields compared to observations when the model is run at eddy resolving resolutions (Wang et al., 2008). Therefore, biharmonic viscosity is used in the momentum equations. Here, $A_h$ is scaled by the cube of the element size with a reference value $A_{h0}$ of $2.7 \times 10^{13}$ $m^4/s$ (Table A1), which is set for the reference resolution of 1 degree.

The continuity equation is used to diagnose the vertical velocity $w$:

$$\partial_z w = -\nabla \cdot \boldsymbol{u} \tag{A2}$$

and the hydrostatic equation is:

$$\partial_z p = -g\rho \tag{A3}$$

where $\rho$ is the deviation from the mean density $\rho_0$.

Tracer equations (A4) and (A5)

$$\partial_t T + \boldsymbol{v} \cdot \nabla_3 T - \nabla \cdot K_h \nabla T - \partial_z K_v \partial_z T = 0 \tag{A4}$$

$$\partial_t S + \boldsymbol{v} \cdot \nabla_3 S - \nabla \cdot K_h \nabla S - \partial_z K_v \partial_z S = 0 \tag{A5}$$

are solved for the potential temperature, $T$, and salinity, $S$ where $K_h$ and $K_v$ are the horizontal and vertical eddy diffusivities, respectively. Laplacian diffusivity is used for the tracer equations. $K_h$ is again scaled as $A_h$ but by the element size with a reference value of $2.0 \times 10^3$ $m^2/s$. These values are set following the convergence study of Wallcraft et al. (2005).





The Pacanowski and Philander (1981) (PP) parametrization scheme is used for vertical mixing, with a background vertical viscosity of $10^{-5}$ m²/s for momentum and diffusivity of $10^{-6}$ m²/s for tracers. The maximum value is set to 0.005 m²/s.

The density anomaly $\rho$ is computed by the full equation of state (A6).

$$\rho = \rho(T, S, p) \tag{A6}$$

The surface and bottom momentum boundary conditions are, respectively:

$$A_v \partial_z \boldsymbol{u} = \tau \tag{A7}$$

$$A_v \partial_z \boldsymbol{u} + A_h \nabla H \cdot \nabla \boldsymbol{u} = C_d \boldsymbol{u} \mid \boldsymbol{u} \mid \tag{A8}$$

where $\tau$ and $C_d$ are the wind stress and the surface drag coefficient, respectively.

The surface kinematic boundary condition is:

$$w = \partial_t \eta + \boldsymbol{u} \cdot \nabla \eta + (E - P - R) + W_{corr} \tag{A9}$$

where $E$ (m/s), $P$ (m/s) are evaporation and precipitation, respectively. $R$ $(km^3/yr)$ is runoff and converted to $m^3/s$ before it is normalised by the area of the buffer zone in the Black Sea (see Fig. 1). Finally, $W_{corr}$ is a correction applied to conserve the volume of the model, as described later.

The sea surface height equation can now be derived from equations A2 and A9 as:

$$\partial_t \eta + \nabla \cdot \int\limits_{z=-H}^{z=\eta} \boldsymbol{u} dz = -(E - P - R) - W_{corr} \tag{A10}$$

The upper limit of integration in (A10) is set to $\eta$ in this version of FESOM and is different from Wang et al. (2008) to provide a non-linear free surface solution.

The bottom boundary condition for the temperature and salinity are

$$(\nabla T, \partial_z T) \cdot \mathbf{n}_3 = 0 \tag{A11}$$

$$(\nabla S, \partial_z S) \cdot \mathbf{n}_3 = 0 \tag{A12}$$

where $\mathbf{n}_3$ is the 3D unit vector normal to the respective surface.

The surface boundary condition for temperature is

$$K_v \partial_z T \big|_{z=\eta} = \frac{Q}{\rho_0 C_p} \tag{A13}$$





where $C_p = 4000 J/(kg\ K)$ and Q $(W/m^2)$ is the surface net heat flux into the ocean.

In global applications, surface salinity is generally relaxed to a climatology to prevent a drift. In our regional application, the water flux term in the boundary condition (A14) is applied over the whole domain whereas the relaxation term is prescribed only in the Black Sea buffer zone.

5  $$K_v \partial_z S \big|_{z=\eta} = S_0(E - P - R) + \gamma(S^* - S_0) - S_{corr} \tag{A14}$$

In the boundary condition (A14), $S_0$ and $S^*$ are the surface salinity and the reference salinity, respectively, $\gamma$ is the relaxation coefficient (Table A1). Finally, $S_{corr}$ is the counterpart of $W_{corr}$ for salinity conservation corresponding to boundary conditions (A14), which will be defined in the following section.

| PARAMETER | DESCRIPTION | VALUE | UNIT |
|---|---|---|---|
| $A_D$ | Model Domain Area | $1.52 \times 10^{11}$ | $m^2$ |
| $A_B$ | Black Sea Buffer Zone Area | $2.26 \times 10^{10}$ | $m^2$ |
| $R_B$ | Black Sea Runoff | Table 1 | |
| $S_0$ | Sea Surface Salinity | | psu |
| $S^*$ | Salinity relaxed in the Black Sea Buffer Zone | Table 1 | psu |
| $\gamma$ | Salinity relaxation coefficient | $5.79 \times 10^{-6}$ | $m/s$ |
| $W_{corr}$ | Water flux correction | | $m/s$ |
| $S_{corr}$ | Salinity flux correction | | $psu\ m/s$ |
| $A_{h0}$ | Horizontal eddy viscosity reference value | $2.7 \times 10^{13}$ | $m^4/s$ |
| $K_{h0}$ | Horizontal eddy diffusivity reference value | $2.0 \times 10^3$ | $m^2/s$ |
| $A_{v0}$ | Vertical background viscosity | $1.0 \times 10^{-5}$ | $m^2/s$ |
| $K_{v0}$ | Vertical background diffusivity | $1.0 \times 10^{-6}$ | $m^2/s$ |

**Table A1.** Parameters used in the model equations, surface boundary conditions and budget corrections.

**Appendix B: Salt Conservation Properties**

10  As our model domain is closed we need to enforce salt conservation. Volume salinity conservation requires the time rate of the change in the volume salinity term in equation (B1) to be zero. A balance must be satisfied between the two integrals.

$$\frac{\partial}{\partial t} \iiint_V S dV = \iint_{A_D} (K_v \partial_z S \big|_{z=\eta}) dA_D = 0 \tag{B1}$$

In FESOM, this balance is achieved by applying a correction for each term separately. The amount of water flux by evaporation, precipitation and runoff is integrated over the surface with every time step (Equation B2). After normalising by the




domain area as in (B3), the surplus or deficit is added to or subtracted from the total water flux equally from each node of the mesh with the $W_{corr}$ terms in equations (A9) and (A10).

$$\Delta_{(E-P-R)} = \int_{x,y} (E - P - R)dxdy \tag{B2}$$

$$W_{corr} = \frac{\Delta_{E-P-R}}{A_D} \tag{B3}$$

The salinity flux is corrected in a similar manner for the boundary condition

$$\Delta S = \int_{x,y} (S_0(E - P - R) + \gamma(S^* - S_0))dxdy \tag{B4}$$

$$S_{corr} = \frac{\Delta S}{A_D} \tag{B5}$$

After applying these corrections, we get a surplus of water corresponding to about 1 mm of sea surface height increase a year, which we believe is due to random numerical errors. Correspondingly, the volume-mean salinity decreases by an order of $10^{-5}$ psu a year. Although these errors may be significant in climate scales, they are acceptable for our six-year long experiment.

*Acknowledgements.* The simulation in this study are performed in CMCC/ATHENA cluster. The work is a part of the PhD study of Ali Aydoğdu and funded by Ca Foscari University of Venice and CMCC. The preparation of this manuscript continued under the funding of
the REDDA project of the Norwegian Research Council during his post-doctoral research. NCAR is sponsored by the US National Science Foundation.



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
