# Peer review of "Circulation of the Turkish Straits System under interannual atmospheric forcing"

_Ocean Science, 2018_

## Referee Comment (RC1) · Anonymous Referee #1 · 5 Mar 2018

The manuscript is a generally well-written study about the Turkish Strait system. The manuscript has potential, because it completes some other studies and provides an overview on the general circulation in the Marmara Sea. However, it has some short-comings hat have to be addressed before the manuscript can be published.

General comments:

How is the wind stress and pressure gradient in the BS dealt with? If you do not have the whole basin, the influence of these forces might not be represented well in the area that is part of your domain.

The boundary conditions are prescribed in a strange way. Authors claim to use a buffer

zone to relax salinity, but then they use a time scale of 2 days. This is, for a large basin like this, basically equal to imposing the salinity.

Validation is not strict. Authors show that they get the orders of magnitude right. But this is not a real validation.

Authors talk a lot about wind forcing, but it is not clear if pressure gradients are also accounted for. Authors also use a correction term in order to conserve mass and salinity. However, as I have understood, they applied this correction term at every time step. This means they do not allow for variations in mass and salinity in the basin. However, I guess that the Danube discharge will lead to a variation of these parameters. So, these variations are completely suppressed. I think the authors will have to come up with another scheme that allows for variations and corrections are imposed in the long run.

The biggest problem for me is with table 4. If I check the difference between ingoing and outgoing fluxes at the Bosphorus, I get a volume difference that corresponds to about 8 meter of water level per year. This clearly cannot be. Authors should check their data and carefully check the volume balance.

Specific comments:

1,3 interface where?

3,22 differ from what

Fig 1 does this show the whole model domain?

5,14-16 how is the salinity of BS and Med kept on a constant level?

5,24 is evaporation considered?

Table 1 please define R and S

7,3 runoff is imposed ,right? Salinity is relaxed in the buffer zone? Why not prescribe

it on the boundary? A relaxation time of two days is practically equal to imposing this value. By the way, what about the other closed boundary, the Mediterranean? How has this boundary been handled?

Fig 3 the units for water fluxes need clarification. They are probably m**3/s per m**2, but this might not be obvious for the reader (it was not for me).

8,1 Q_H is positive when? From atmosphere to ocean?

11,4-7 I would like to point out the problem with T after 6 years of integration in deeper layers. As can be seen from figure 9 the lower layers have warmed up by nearly 2 degrees. What is the explanation for this behavior?

Fig 11 when you discuss wind forcing, you should also discuss atmospheric pressure forcing. What about pressure gradients over the same period? I guess pressure is included in the model equations, right?

13,1-2 wind forcing is not the only responsible for sea level fluctuations. Did you look in the pressure variations?

13,7 why excluding pressure forcing?

Table 4 The net flow should be really exactly the same for the northern and southern location. If I assume the length of the Bosphorus with 30 km and its average width with 3 km, I get an area of 100 km**2. The net flux difference between northern and southern sections is 0.8 km**3. This would correspond to an 8 m water level difference over one year. This clearly cannot be true. So what is wrong with this calculation? The way you compute fluxes, or your numbers? Please clarify.

17,5 also normalized by the density

17,10 eq. 3 has no right hand side...

18,1 is this total kinetic energy, or only the one caused by wind effects?

19,8 ?????

Appendix:

A1 diffusion term. Sure about the 4th derivative in the term with A_h? Ok, you use bi-harmonic diffusion.

24,2 what about unstable stratification? Can you resolve problems arising with this small vertical diffusivity?

24,13 what does it mean, normalized to the buffer zone?

26,10 I do not understand: you compute the correction term at every time step? So you do not allow the model to become more or less saline during special events (spring river run off, etc.)? I would have done this computation on a mean (maybe annual) value. You clearly do not want a drift of salinity, but variations should be allowed
* * *

---

## Referee Comment (RC2) · Anonymous Referee #2 · 11 Mar 2018

Authors use one model with a very good record (FESOM) and apply it to a unique ocean region, the Marmara Sea. I find the dynamics of Marmara Sea a novel topic. It is oceanographically interesting because circulation in the Marmara Sea is strongly dependent on the exchange with the neighboring basins, the Black Sea and the Mediterranean. However, I find some fundamental problems with this manuscript exactly in this part and do not recommend publishing it. With the following comments, I want to elucidate the basic problems, and help authors sharpen their paper if they want to submit a new manuscript. 1. What is strongly needed is that authors a) Demonstrate the power of using an unstructured-grid model compared to structured grid models when addressing the dynamics as dependent on the transport in the Straits of Bosporus and

[Figure]

Marmara. There are some references to earlier works, but there is not a critical comparison with structured grid models. Is the skill of unstructured-grid model and the proposed setup better in comparison to what is known from earlier works on the Marmara Sea modelling? b) Clearly demonstrate the superiority of FESOM compared to other unstructured grid models in the present study. Critical statements in the introduction about possible problems in other unstructured grid models need to be supported by a deeper analysis of results and inter-comparisons. Advantages or drawbacks of explicit and implicit models have to be made clear, in particular the representation of dominant processes in the studied area by different approaches. This would increase the credibility of present results; otherwise criticism would not be justified. c) Analyze, in a quantitative way, processes in the straits and the skill of model to replicate the basic physics. There your model is superior in comparison with the structured grid ones and you need to demonstrate this.

2. One sea forced by two straits presents a very interesting system to explore salt and mass balances and the role of straits for the water mass formation, in particular. This issue is only marginally addressed. Analysis is not very symmetric; more attention has been given to the Bosporus. One would like to see a figure similar to Fig. 12 for the Dardanelles. This is very important because the latter provides the source of deep water masses (see Fig. 6b; why is this figure cut at 100m?). The analysis of Fig 6 and associated processes needs to be extended down to the depth of the maximum reach of Aegean Sea water. It could be that the trend in Fig. 7 reflects a trend in the deep waters (or a problem with initialization). These comments lead me to the conclusion that authors have to deepen the physical interpretation of their results.

3. Some of the presented results could reveal that the mass and salt balance in the model is not correctly represented. This is a fundamental issue, which could convey very negative miss-interpretation of FESOM skills. Net water transport in Bosporus, as seen in Tabl. 4, is ∼150kmˆ3/yr; in the Dardanelles it is ∼100kmˆ3/yr. This is in contradiction with the net transport published earlier by one of the authors (Ozsoy and

Unluata, 1997, their Fig. 5) where it was shown that the water flux at the Marmara air-sea interface is minor in comparison to the straits transport. Results in Table 4 are also in contradiction with the statement "The resulting net water flux P - E varies between -4.7x10−8 and 2.5x10−8 m/s." Taking the area of Marmara Sea ∼11.000 km$^2$ and looking in Fig. 3 where the mean water flux is ∼- 1x10−8 m/s yields also a negligible water flux at the Marmara Sea surface. Authors have to look closely how to explain the difference between ∼150km^3/yr and ∼100km^3/yr. They have to carefully check the conservation of mass and tracers and include this, if they submit a new manuscript. Unlike the models with large open boundaries, the Marmara Sea gives unique opportunities to address conservation properties and authors have to take advantage of it.

4. A further problem is identified in the comparison between Table 4 and Table 5 demonstrating that water flowing through the strait of Bosporus is two times less than what is reported in the literature (∼300 km^3/yr). There are two problems here.

a) I wonder how with ∼two times smaller net transport authors simulate realistically the two layer transport and its impact on the Maramara Sea circulation. What about the deep layer transport in the Dardanelles? Isn't there a trend in the system if you have unrealistic fluxes in the straits (see Fig. 7b)? b) Two times smaller net transport means that the fresh water balance in the Black sea is wrong. Led by these arguments, I again propose that authors present clearly the model forcing at all open boundaries, rivers and air-sea interface, as well as and the corresponding fresh water and salt balances for the Black Sea, Aegean Sea and Marmara Sea.

---

## Short Comment (SC1) · 11 Mar 2018

I read the paper of Aydoğdu and co-authors with great interest. I find some problems in the interpretation of previous results

1. The criticism of the paper of Stanev et al. (2017) is mostly off the mark. Below are their texts: ' Stanev et al. (2017) used an implicit advection scheme for transport to handle a wide range of Courant numbers (Zhang et al., 2016) while satisfying the stability of the solution. However, the computational burden of using an implicit scheme imposed a coarser model resolution with 53 vertical levels at the deepest point of the Black Sea. I note that this limitation may, particularly in the Bosphorus, lead to excessive vertical mixing or a widened interface thickness, which are crucial for the intrusion of the Mediterranean origin water into the Black Sea. This can be seen in their Figures 11c and 11d, for example.'

I want to bring to the attention of authors that: a) 53 vertical levels in the Bosporus, which in some places is ∼30 m deep, is less than 1 m vertical resolution with the used vertical coordinates, that is better than the resolution used by authors. They say "The vertical resolution is 1 m in the first 50 m". The conclusion is that authors have to spend some time to reading carefully what other scientist have published. b) SCHISM uses explicit Eulerian-Lagrangian approach for momentum advection (which is unconditionally stable). It also uses an implicit scheme for terms in the momentum and continuity equations that place most stability constraints (pressure gradient, divergence, vertical viscosity). Most importantly, the size of matrix from the implicit scheme is determined by number of horizontal nodes, not this times number of levels. In fact we have used 92 levels in one version of our Kuroshio simulation and it went fine. So it's not a fundamental problem for us to use a large number of levels - it's just a practical consideration given our limited computational resource (see also my comment (a)). In any case, your results inspired us to try larger number of levels in the future.

Our experience with our own and other Z-coordinate models is that it requires a large number of levels to get reasonable stratification and bottom intrusion right (due to staircase). We switched to hybrid coordinates precisely because of this. I hope that authors will avoid conveying this kind of mis-information.

2. I expected, after having seen their wrong criticism against the work of Stanev et al. (2017), to see some examples. Unfortunately, this paper is only cited in the introduction, so their statement there is totally unjustified. A fair approach would be to clearly demonstrate what their progress is in comparison with earlier unstructured-grid experiments.

Joseph Zhang March 2018

---

## Referee Comment (RC3) · Anonymous Referee #3 · 20 Mar 2018

The paper presents a valuable contribution to the understanding of the circulation in the Turkish Straits System, connecting Black Sea and Mediterranean Sea through Bosporus Strait, Marmara Sea and Dardanelles Strait. In fact, the focus is put on the Marmara Sea circulation and the heat and mass transport through Bosphorus, to a less extent in the Dardanelles Strait. It deals with the physical modeling of the TSS, thus it lies within the scope of Ocean Science.

The Introduction section gives very good overview of the related studies in the region. The used model FESOM is unstructured grid model, set up with 110 vertical levels and horizontal resolution ranging from 65 m in Bosphorus and 150 m in Dardanelles

to 1.5 km in Marmara Sea and 5 km in the Black Sea. There is a similar study with unstructured grid model for the same region of Stanev et al. 2017, focusing on the Black Sea dynamics.

The numerical simulation covers 6 years with realistic atmosphere forcing and climatological fresh water flux forcing at the open boundary. There is a reference to the initial implementation of Gürses et al. 2016, but the authors might give some details on it.

The results for the water mass structure in Marmata Sea and the straits are in general interesting and worth publishing. However, it is necessary to discuss the result from the validation, showing that the model error increases in time (Table 2). From Figure 7 it is seen the negative trend in the volume temperature and positive for the volume salinity, what could be the reason for this trend?

The section on the sea level and mass transport in the straits is also interesting. Figure 11 does not quite clear show what is the correlation between observations and simulations. There is information on the lower and upper layer velocity, but it is advisable to give more details on the variability of the interface depth. The chapter on Marmara Sea dynamics is well written. Figure 15 and 17 might look better in color as Figure 16.

Some concrete remarks:

-The terms for $\alpha$ and $\beta$ are thermal expansion and haline contraction coefficients, their values might be also given in the table with parameters.

-The Figure 3 could be also in colors, now it is not very well read.

-Figure 6b shows cooling on the surface at the south end of Bosphorus, is it realistic?

-There is an unfinished sentence with ??????

---

## Short Comment (SC2) · 22 Mar 2018

To correctly reproduce the exchange of water and energy between adjacent seas, there is the need to jointly describe the general circulation at the basin scale and the processes in the straits, detectable at very fine spatial and temporal scales. The paper of AydoĂ§du et al. addresses this issue by applying an unstructured numerical model to a unique domain representing the Turkish Strait System, part of the Black Sea and part of the Aegean Sea. Similarly, in Ferrarin et al. 2018, (Ferrarin, C., D. Bellafiore, G. Sannino, M. Bajo, and G. Umgiesser. 2018. Tidal dynamics in the inter-connected Mediterranean, Marmara, Black and Azov seas, Prog. Oceanogr., 161, 102–115, doi:

10.1016/j.pocean.2018.02.006), with the use of the unstructured SHYFEM model we simulated the tidal propagation and transformation in the system of inter-connected basins formed by the Mediterranean, the Marmara, the Black and the Azov seas.

The numerical model applied by Aydoğdu et al. to the TSS was properly described, even if some additional details about the forcing and boundary conditions are needed.

My major concern with this study is on the capability of the present model application to correctly reproduce the water circulation in the system and the exchange dynamics at the Dardanelles and Bosphorus straits. The authors argue in the abstract and in the results description that the simulation maintains its realism. However, the evidences provided in the paper demonstrate that some numerical model results are unrealistic. In particular:

- The water fluxes presented in the table 4 are about half than what is reported in the literature (Table 5).

- The water fluxes presented in Figure 13 do not reproduce the observed variability and magnitude.

- The thermocline and halocline are generally deeper that what is observed (Figure 9).

- The numerical model seams not able to correctly reproduce the observed sea level differences between Yalova and Sile. The authors should provide statistics (RMSE, BIAS, R2) of the model performance for the water levels.

Concluding, to my opinion, the authors have to provide clear and robust calibration and validation of their numerical model application before inferring on the water circulation of the Turkish Strait System.

---

## Author Comment (AC1) · 29 May 2018

The comment was uploaded in the form of a supplement:
https://www.ocean-sci-discuss.net/os-2018-7/os-2018-7-AC1-supplement.pdf

---

## Author Response (AR1)

**"Circulation of the Turkish Straits System under interannual atmospheric forcing"**

**Aydoğdu et al., 2018, in revision, submitted to Ocean Science**

**General response to all the reviewers**

We are grateful to the anonymous referees for their careful assessment and Dr. Zhang and Dr. Ferrarin for their short comments on the original manuscript of our study on the circulation of the Turkish Straits System. Their comments and suggestions have helped us to substantially improve the manuscript. Below are our point-by-point responses to their comments, which should resolve all points raised by the referees towards publication of our work. Our proposed modifications in line with the present responses are enclosed at the end of this document in a marked-up version of the manuscript, where the figure and table numbering used in the following responses are referenced to the marked-up version. The reviewers' comments and our responses are in italic and normal fonts, respectively, throughout this document, whereas the captions are given in bold fonts.

**Author' response to RC#1**

*The manuscript is a generally well-written study about the Turkish Strait system. The manuscript has potential, because it completes some other studies and provides an overview on the general circulation in the Marmara Sea. However, it has some shortcomings that have to be addressed before the manuscript can be published.*

**General comments**
*How is the wind stress and pressure gradient in the BS dealt with? If you do not have the whole basin, the influence of these forces might not be represented well in the area that is part of your domain.*

We impose wind stress over the entire model area, following the earlier development in the Mediterranean Sea making use of a closed Atlantic box similar to ours (Tonani et al., 2008), which only has been configured to allow to understand the main wind-driven circulation characteristics. We do not apply atmospheric pressure forcing, because that would require having proper set of open lateral boundary conditions in the Aegean and Black Sea boxes with a potential to excite basin-modes in the adjacent basins that we initially try to exclude from our simulations.

*The boundary conditions are prescribed in a strange way. Authors claim to use a buffer zone to relax salinity, but then they use a time scale of 2 days. This is, for a large basin like this, basically equal to imposing the salinity.*

We indeed impose a monthly climatology for salinity in the buffer zone with a short relaxation time because we want to fix strictly the salinity value in the buffer zone to adequately represent major freshwater effects that should feed back to the TSS region where our main interest is concentrated.

*Validation is not strict. Authors show that they get the orders of magnitude right. But this is not a real validation.*

The observational data at the scale of the Marmara Sea is scarce and to our knowledge we have made use of all the data available for a basin-scale validation of the T, S properties. In addition, we have compared the Bosporus Strait with the available current data of Jarosz et al. (2011, 2012) during the experiment period. Table 2 shows the quantitative estimates of the RMSE for temperature and salinity. Moreover, the sea level in the Marmara Sea is quantitatively compared with a four-year time series of observations obtained from tide gauges.

*Authors talk a lot about wind forcing, but it is not clear if pressure gradients are also accounted for.*

We do not consider atmospheric pressure, since we limit our attention to wind-driven part of the circulation as already stated. We now have explicitly commented this in Appendix A, where the model equations are presented. We also changed our title to read as:

"Circulation of the Turkish Straits System under interannual atmospheric forcing"

*Authors also use a correction term in order to conserve mass and salinity. However, as I have understood, they applied this correction term at every time step. This means they do not allow for variations in mass and salinity in the basin. However, I guess that the Danube discharge will lead to a variation of these parameters. So, these variations are completely suppressed. I think the authors will have to come up with another scheme that allows for variations and corrections are imposed in the long run.*

The total volume of the model needs to be conserved since the basin is closed, while also to counter resultant climate drift. We prefer to keep the volume to remain unchanged at every time step because that is the simplest choice in the absence of *a priori* information on how secular variations of the various flux components are balanced on climatological time scales. In order to have the basin averaged sea level tendency to vanish, we need to correct the surface boundary conditions for the vertical velocity to have zero mean vertical velocity.

Because of the dominating influence of the fresh-water term R in the Black Sea, the average of the net water flux (E-P-R) is not zero. We therefore needed to correct the vertical velocity, as applied by Tonani et al. (2008) for regional domains or by Gent et al. (1998) for the global ocean.

The reasoning is the same for salinity. We conserve the total salinity strictly, however, this does not prevent spatial variation of salinity since the correction is appied to the surface layer equally.

*The biggest problem for me is with table 4. If I check the difference between ingoing and outgoing fluxes at the Bosphorus, I get a volume difference that corresponds to about 8 meter of water level per year. This clearly cannot be. Authors should check their data and carefully check the volume balance.*

We gratefully acknowledge the reviewer's close attention to volume transport calculated by the model. In addition, extensive discussion is now provided in relation to the volume transport issues brought to

our attention by the above comments and also by comment #3 of reviewer #2. The calculation of fluxes was indeed not very accurate and now they are corrected in Table 4 as well as in Fig. 14.

The major problem in the computation of volume transport was that we used daily snapshot velocities instead of daily mean velocities. We also had some inaccuracies in the post-processing computations of transport on the unstructured grid. Both sources of inaccuracy have now been eliminated to yield correct transport estimates. The net transport difference between the northern and southern sections of the Bosphorus Strait is now computed as 0.5 km$^3$/yr, with the higher value in the northern section.

On the other hand, the reduction of transport as one moves from north to south along the TSS is a direct result of our present correction scheme necessitated by volume conservation in the model domain with closed lateral boundaries. As applied here, the correction scheme results in a continuous decrease of transport along the way from the Black Sea to the Aegean Sea by extracting water from the surface to balance the runoff water flux R introduced at the buffer zone minus the specified surface flux E-P up to the same point as discussed in Appendices A and B. We further discuss the effects of surface flux corrections on the Marmara Sea in our response to reviewer #2. We refer to sections 3.1 and 3.3 in the marked-up version of the manuscript respectively for the surface water fluxes and the resulting volume transports.

**Specific comments**

*1,3 interface where?*

We remove the phrase 'interface layer' and use 'pycnocline'. The sentence now reads as: "The depth of the pycnocline between the upper and lower layers remains stationary after six years of integration,..."

*3,22 differ from what*

The sentence is clarified as "… differ significantly in each of their experiments with ..."

*Fig 1 does this show the whole model domain?*

This is the whole model domain. We clarify it in the caption.

*5,14-16 how is the salinity of BS and Med kept on a constant level?*

The salinity in the model domain evolves throughout the simulation according to water and heat fluxes from the atmosphere and the river runoff flux supplied through the Black Sea buffer zone. Therefore, it varies spatially in the model domain. The salinity in the Black and Mediterranean Seas are not kept constant but the volume salinity is conserved over the whole model domain by the correction term in Appendix B.

*5,24 is evaporation considered?*

Yes, and it is shown in Fig. 3 for the Marmara Sea. We now have added a similar figure for the whole

domain with units for the water fluxes converted to km$^3$/yr in order to extend our discussion on volume fluxes in the Marmara Sea and the Straits. Related section 3.1 is modified accordingly.

*Table 1 please define R and S*

We have included the notation R after "river discharges" and S* after "salinity relaxation" in parenthesis in the caption.

*7,3 runoff is imposed, right? Salinity is relaxed in the buffer zone? Why not prescribe it on the boundary? A relaxation time of two days is practically equal to imposing this value. By the way, what about the other closed boundary, the Mediterranean? How has this boundary been handled?*

In order to prescribe an "entering volume flux" in the fully closed model basin and properly account for mass conservation, we imposed the vertical velocity boundary condition (A9). The Black Sea river runoff is distributed at the surface in the Black Sea buffer zone at the furthest limit of the model area excluding the actual major rivers. Moreover, we tightly relaxed salinity to monthly climatology values in the same box to keep realistic values of salinity entering in the Bosporus.

In the simulation presented in the paper, we only have a buffer zone in the Black Sea. However, earlier tests using an additional buffer zone near the Aegean southern boundary were presented by Gürses et al. (2016), producing comparable results.

*Fig 3 the units for water fluxes need clarification. They are probably m\*\*3/s per m\*\*2, but this might not be obvious for the reader (it was not for me).*

We now present the water fluxes in km$^3$/yr in Fig. 3 to be comparable with the units of the volume transport through the straits.

*8,1 Q_H is positive when? From atmosphere to ocean?*

Correct. We add it to the corresponding sentence as "...Q_H is the heat flux (positive towards the ocean)..."

*11,4-7 I would like to point out the problem with T after 6 years of integration in deeper layers. As can be seen from figure 9 the lower layers have warmed up by nearly 2 degrees. What is the explanation for this behavior?*

If we understood correctly, the reviewer points out the difference between the observed and simulated temperature in the lower layer. We know that the lower layer water masses take several years to evolve after initialization, as can be seen in the trend in Fig.7. On the other hand, we also note that the observations in June 2013 are 1°C cooler than the ones in April and October 2008 although the 2013 summer measurements normally are expected to be warmer, possibly indicating such interannual changes in the sub-halocline water. The physical reason behind the observed variability could be the strong cooling event in 2012 (e.g. Benetazzo et al. 2014), which is not reproduced by the model; a side-effect that should be further investigated in the future.

*Fig 11 when you discuss wind forcing, you should also discuss atmospheric pressure forcing. What about pressure gradients over the same period? I guess pressure is included in the model equations, right?*

We do not investigate the short-term atmospheric pressure effects investigated because we only study the TSS circulation developed under the effects of the surface and lateral fluxes of water, heat and salt in this paper.

*13,1-2 wind forcing is not the only responsible for sea level fluctuations. Did you look in the pressure variations?*

Again, we do not have the atmospheric pressure included at the moment.

*13,7 why excluding pressure forcing?*

In the present study, we are mainly interested in the circulation developed in the TSS region, i.e. the two Straits and the Marmara Sea, focusing on the impacts of the wind forcing as well as water and heat fluxes on a regional scale. The main reasons to exclude the pressure forcing were to limit unintended feedback of motions that could independently develop in the falsified (cut-down) geometries of the Aegean and Black Sea boxes, including unrealistic basin modes that could be excited by the large scale atmospheric pressure field in these closed basin reservoirs. We believe that to adequately represent atmospheric pressure effects on a small dimension regional model, one needs to have open boundary conditions applied on a realistic domain.

*Table 4 The net flow should be really exactly the same for the northern and southern location. If I assume the length of the Bosphorus with 30 km and its average width with 3 km, I get an area of 100 km\*\*2. The net flux difference between northern and southern sections is 0.8 km\*\*3. This would correspond to an 8 m water level difference over one year. This clearly cannot be true. So what is wrong with this calculation? The way you compute fluxes, or your numbers? Please clarify.*

We responded to this question of the reviewer in the general comments. The issue related to volume transports is discussed extensively also in response to the comment #3 of the reviewer #2.

*17,5 also normalized by the density*

We added "normalized by density" in the figure caption and text.

*17,10 eq. 3 has no right hand side. . .*

Thanks. We rephrase the sentence as " ...we normalize the integral in (3) by the volume..."

*18,1 is this total kinetic energy, or only the one caused by wind effects?*

This is total kinetic energy. We remove the word "resulting" to avoid confusion.

*19,8 ?????*

It is a latex related typo. Question marks are removed.

*Appendix:*

*A1 diffusion term. Sure about the 4th derivative in the term with A_h? Ok, you use bi-harmonic diffusion.*

Correct.

*24,2 what about unstable stratification? Can you resolve problems arising with this small vertical diffusivity?*

The "background" vertical viscosity and diffusivity are minimum thresholds. Typically, the spatially varying vertical viscosity and the diffusivity are approximately two orders of magnitude greater than the background values. Therefore, there is no stability issue arising due to small vertical diffusivity.

*24,13 what does it mean, normalized to the buffer zone?*

That simply means that the water volume flux is divided by the buffer zone area to distribute it equally in the buffer zone in units of vertical velocity. We have rephrased the sentence.

*26,10 I do not understand: you compute the correction term at every time step? So you do not allow the model to become more or less saline during special events (spring river run off, etc.)? I would have done this computation on a mean (maybe annual) value. You clearly do not want a drift of salinity, but variations should be allowed*

We compute the correction terms W_corr and S_corr at every time step because we insist to conserve the mass and the volume at all times in the closed basin configuration. This requirement could in fact be released in the future if a priori information on all the water flux components, e.g. if the riverine, Bosphorus and surface components and their T, S characteristics had existed or alternatively, open boundary conditions could be accurately specified. In the present case of closed basin configuration, we prefer to abide by the requirements of mass conservation. This doesn't hinder daily or seasonal variation in the TSS region, which is the area of our main interest. The seasonal cycle in the Marmara Sea can be clearly seen in Fig. 7 even for the volume mean salinity.

**References**

Benetazzo, A., A. Bergamasco, D. Bonaldo, F. Falcieri, M. Sclavo, L. Langone, and S. Carniel(2014). Response of the adriatic sea to an intense cold air outbreak: Dense water dynamics and wave-induced transport. Progress in Oceanography 128, 115 – 138.

Gent, P. R., Bryan, F. O., Danabasoglu, G., Doney, S. C., Holland, W. R., Large, W. G., & McWilliams, J. C. (1998). The NCAR climate system model global ocean component. *Journal of Climate*,*11*(6), 1287-1306.

Tonani, M., Pinardi, N., Dobricic, S., Pujol, I., & Fratianni, C. (2008). A high-resolution free-surface model of

the Mediterranean Sea. *Ocean Science*, *4*(1), 1-14.

**Author' response to RC#2**

Authors use one model with a very good record (FESOM) and apply it to a unique ocean region, the Marmara Sea. I find the dynamics of Marmara Sea a novel topic. It is oceanographically interesting because circulation in the Marmara Sea is strongly dependent on the exchange with the neighboring basins, the Black Sea and the Mediterranean. However, I find some fundamental problems with this manuscript exactly in this part and do not recommend publishing it. With the following comments, I want to elucidate the basic problems, and help authors sharpen their paper if they want to submit a new manuscript.

**Major comments**

*1. What is strongly needed is that authors*

a) *Demonstrate the power of using an unstructured-grid model compared to structured grid models when addressing the dynamics as dependent on the transport in the Straits of Bosporus and Marmara. There are some references to earlier works, but there is not a critical comparison with structured grid models. Is the skill of unstructured-grid model and the proposed setup better in comparison to what is known from earlier works on the Marmara Sea modelling*

To our knowledge, the only structured grid model implementation in the whole Turkish Straits System (TSS) is the MITgcm configuration of Sannino et al. (2017) that used a curvilinear grid resolving the straits with higher resolution compared to the neighbouring basins. Other structured grid model implementations are used to simulate only individual components of the TSS, and not the full TSS as we intend to fully resolve fine geometries of fine scale flow features such as the two narrow straits in the present paper. On the other hand, there may be other ongoing attempts aiming to study inter-basin interactions by coarse structured grids facing immense computing requirements could not possibly resolve the complex, narrow straits and their adjacent regions to converge to realistic results. We have not attempted to review such cases as they would be outside our area of interest.

In Sannino et al. (2017), the focus is on the response of the TSS to a range of varying barotropic volume flux, whereas in our study, we focus on the impact of atmospheric forcing on the circulation in the Marmara Sea. Therefore, it is not possible to compare directly the two studies. However, we tried to identify some links between the simulated circulation in both studies in sections 3.3 and 3.4 as well as in the summary and discussion sections of the manuscript.

b) *Clearly demonstrate the superiority of FESOM compared to other unstructured grid models in the present study. Critical statements in the introduction about possible problems in other unstructured grid models need to be supported by a deeper analysis of results and inter-comparisons. Advantages or drawbacks of explicit and implicit models have to be made clear, in particular the representation of dominant processes in the studied area by different approaches. This would increase the credibility of present results; otherwise criticism would not be justified.*

The only other unstructured grid modelling study investigating the TSS together with the Black Sea by Stanev et al. (2017) considers the entire region in the same model domain, but both the resolution and the number of computational nodes are essentially much less than ours, the topography also being not well represented. The study manages to arrive to few results that seem to be in line with our own conclusions on the coupled behaviour of the TSS, although the focus of the paper is more on the intrusion of the Bosphorus inflow into the Black Sea, rather than the specifics of the TSS. Therefore there is not much to be discussed with respect to specific features of the models.

A further unstructured grid model was recently presented by Ferrarin et al. (2018) after our submission of the present paper, studying the tidal dynamics in the inter-connected Black Sea – Mediterranean Sea system. We now mention their study in the introduction section, although there is no possibility of inter-comparison.

c) *Analyze, in a quantitative way, processes in the straits and the skill of model to replicate the basic physics. There your model is superior in comparison with the structured grid ones and you need to demonstrate this.*

The reviewer is quite right in stating the superiority of the present model. The absolute requirement to represent extreme physical processes of the TSS at the finest scale possible in a model of the region has been the main motivation on our part to select a full-featured hydrodynamic model of unstructured genre. In addition, the need to represent all of the possible strait processes in full had been demonstrated by earlier studies by Sözer and Özsoy (2017) and the need to represent the straits as well as the Marmara Sea and the adjacent basins in fine detail in a model of the whole region had been found to have a central importance in Sannino et al. (2017). Essentially, it has been shown that the 'systems behaviour' of the model of the whole TSS behaves much differently from a linear extension of the results from models of its individual parts. Despite these considerations of dissimilarity, we further qualitatively compare our results with Sözer and Özsoy (2017) in section 3.3 to show that the main features found in the stand-alone strait model are adequately represented in normal and extreme situations. Perhaps the only missing part is the detailed comparison with the behaviour of the Dardanelles Strait, although that probably has to wait for a full-fledged investigation of the strait hydrodynamics by a stand-alone model. These motivations and the main results of former studies as applied to the present study have been quite transparently discussed in the introduction section and throughout the paper.

2. *One sea forced by two straits presents a very interesting system to explore salt and mass balances and the role of straits for the water mass formation, in particular. This issue is only marginally addressed. Analysis is not very symmetric; more attention has been given to the Bosporus. One would like to see a figure similar to Fig. 12 for the Dardanelles. This is very important because the latter provides the source of deep water masses (see Fig. 6b; why is this figure cut at 100m?). The analysis of Fig 6 and associated processes needs to be extended down to the depth of the maximum reach of Aegean Sea water. It could be that the trend in Fig. 7 reflects a trend in the deep waters (or a problem with initialization). These comments lead me to the conclusion that authors have to deepen the physical interpretation of their results.*

As this paper is not dedicated to water mass formation analysis but to a novel understanding of the role played by atmospheric and water fluxes forcing on the TSS circulation, we only interpret what we observe in the model results as regards water masses and prefer to reserve to address details of water mass formation processes for future studies. We should also like to express our private view on the practicality of such analysis at this stage: the rather demanding nature of the model runs and the post-processing in terms of high performance computing and storage capacity has been a deterrent to have repeated scenario runs that would be needed for water mass analyses.

The asymmetry in the analysis towards Bosphorus is not only because historically more is known about this Strait, but also because the Bosphorus is the more strictly limiting member of the TSS as a result of its rather special "maximal exchange" hydraulic regime, as demonstrated by recent modelling results by Sözer and Özsoy (2017) and Sannino et al. (2017). The role of the larger Dardanelles Strait is also important, and should be further investigated to reveal its different hydraulic regime and role in the overall system behaviour.

We provide the same figure (now Fig. 13 in the marked-up version) for the Dardanelles as Fig.12 of the Bosphorus and discuss it in section 3.3. Very briefly, the layered-structure of the water column in normal conditions in Nov. 15, 2008 (Fig.13a) resembles the case of Sannino et al. (2017, Fig. 7). In Nov. 22, 2008, as a result of the more active atmospheric conditions, the salinity in the upper layer increases by extensive mixing. Further, the stratification is broken partially in the Marmara Sea side of the strait. The structure of the two-layered flow and response of the model to the atmospheric events seems correct.

We note that the trend in Fig. 7 is due to the initialization. The initial fields are obtained from a lock-exchange simulation started from three different vertical profiles for each of the Black, Marmara and Aegean Seas as explained in section 2. We can deduce that the profiles chosen to initialize Marmara Sea is warm and the simulated volume mean temperature gradually gets cooler until 2012 with the intrusion of the Aegean Sea water. In Fig. 6, we showed only the first 100 m. before since our analysis is focused on the upper layer structures interaction with the atmospheric forcing. However, we now provide the figure extended down to 600 m. We discuss this issue in the section 3.2 of the revised manuscript.

*3. Some of the presented results could reveal that the mass and salt balance in the model is not correctly represented. This is a fundamental issue, which could convey very negative miss-interpretation of FESOM skills. Net water transport in Bosporus, as seen in Tabl. 4, is ~150km^3/yr; in the Dardanelles it is ~100km^3/yr. This is in contradiction with the net transport published earlier by one of the authors (Ozsoy and Unluata, 1997, their Fig. 5) where it was shown that the water flux at the Marmara air- sea interface is minor in comparison to the straits transport. Results in Table 4 are also in contradiction with the statement "The resulting net water flux P - E varies between -4.7x10−8 and 2.5x10−8 m/s." Taking the area of Marmara Sea ~11.000 km² and looking in Fig. 3 where the mean water flux is ~- 1x10−8 m/s yields also a negligible water flux at the Marmara Sea surface. Authors have to look closely how to explain the difference between ~150km^3/yr and ~100km^3/yr. They have to carefully check the conservation of mass and tracers and include this, if they submit a*

*new manuscript. Unlike the models with large open boundaries, the Marmara Sea gives unique opportunities to address conservation properties and authors have to take advantage of it.*

We thank a lot to the reviewer for drawing our attention to volume transports, which appeared inconsistent in the earlier manuscript. A similar issue has been pointed out by reviewer #1 for the Bosphorus Strait. We discuss the problem in full details here, update and provide corrected estimates of fluxes.

The major problem with the volume transport computations was that they were based on daily snapshots rather than daily averages of the current velocity data in the model outputs. We have now corrected and documented the estimates in Table 4 and Fig. 14 (please see the marked-up version of the manuscript below).

Secondly, as our implementation of the TSS model has closed lateral boundaries we needed to have the net water flux over the domain to be zero at each time step. Our choice to satisfy this requirement is to take out the excess water volume flux that arises at every time step and distribute it over the model domain in order to as a surface flux correction. This is also the solution used by Gent et al., (1998) as well as all the global implementations of FESOM referred in the manuscript. (Please see other considerations in the reply to Reviewer #1).

However, one of the drawbacks of our choice is the one pointed out by the reviewer. The volume flux entering into the Straits has already been partly taken out from the surface by $W_{corr}$ term in equation (A9). This explains also why the barotropic fluxes in the Straits are gradually decreasing throughout the TSS from northern Bosphorus to southern Dardanelles sections.

More specifically, Fig.3 which shows the water fluxes in the Marmara Sea now includes also the correction term W_corr in Fig.3b in the revised version. A similar figure for the whole domain is also provided in Fig.3a. For the whole domain, the mean runoff is 287 km$^3$/yr in the Black Sea buffer zone with seasonal variability. The total mean evaporation and precipitation are -131.5 and 70.8 km$^{3/}$/yr, respectively (positive into the ocean). Their sum is 226.4 km$^3$/yr which is the mean water flux correction to conserve volume over the whole domain.

As discussed in the Appendix B, we distribute this correction to the whole surface, weighted by the area of each sub-region (please see Table A which now includes area of different compartments of the TSS). The mean correction applied in the Marmara Sea is, therefore, -15.9 km$^3$/yr (Fig. 3b). The net outflux is -17.1 km$^3$/yr including evaporation minus precipitation.

Given the net volume transports through the southern Bosphorus and northern Dardanelles as -150.0 and -132.8 km$^3$/yr, the difference between the two lateral boundaries of the Marmara Sea is -17.2 km$^3$/yr which is balanced by the surface water flux. The surplus is 0.1 km$^3$/yr, which is less than 0.1% of the volume transport through the southern Bosphorus transect. This small numerical error could originate from the transport computation which approximates the fluxes through the fully-shaved cells at the bottom boundary layer. The transport difference between the two ends of the Dardanelles is

slightly higher possibly because of the increased evaporation at the wide section off the Strait and the deep channel topography on one side of an otherwise flat topography at the exit region also being liable to errors in flux computations. Overall, the errors are negligible and the budgets computed separately for each component of the TSS are closed, verifying continuity in the average transports through the system.

*4. A further problem is identified in the comparison between Table 4 and Table 5 demonstrating that water flowing through the strait of Bosporus is two times less than what is reported in the literature (∼300 km^3/yr). There are two problems here.*

None of the estimates in Table 5, except Kanarska and Maderich (2008), is a numerical simulation. All of the estimations are done using mass conservation by assuming an E-P-R=300 km$^3$/yr in the Black Sea. We provide it to the model as we have shown in our response to the comment #3. However, our choice of correction in our modelling approach reduces the fluxes approximately 50 % and we know how to correct it. We note that modeling study of Kanarska and Maderich (2008) also computes less net transport compared to other estimates which is possible in numerical models. All of the computations are only estimates. Here, in our first try, what is important for us is to show that the  water fluxes are balanced to conserve average properties in the closed domain used. In our further studies, we intend to improve the solution, by applying the water flux correction finally to a water flux sink that takes out the residual water flux after accounting for the total E-P-R in the system and adding to it any corrections tightly balancing the water budget. (see further comments provided in response to Reviewer #1).

*a) I wonder how with ∼two times smaller net transport authors simulate realistically the two layer transport and its impact on the Marmara Sea circulation. What about the deep layer transport in the Dardanelles? Isn't there a trend in the system if you have unrealistic fluxes in the straits (see Fig. 7b)?*

There is no trend in the system as can be seen from Fig. 11 for the sea level and Fig. 13 (now Fig.14 in the marked-up version) for the volume transports. It is the correction term $W_{corr}$ in equation (A9) that reduces the volume transport along the way from the Black Sea to the Aegean Sea. Further care is given to stress the important role of the volume and mass conservation in a closed lateral boundary models. In fact, the purpose of the whole flux correction procedure is to satisfy conservation in the closed basin model, and that strategy has worked, albeit some relatively simple choice, to prevent physically and numerically based trends in the simulations.

*b) Two times smaller net transport means that the fresh water balance in the Black sea is wrong. Led by these arguments, I again propose that authors present clearly the model forcing at all open boundaries, rivers and air-sea interface, as well as and the corresponding fresh water and salt balances for the Black Sea, Aegean Sea and Marmara Sea.*

There is no imbalance or trend due to the water fluxes in the system, we have clarified the role of the correction term and its impact on the net transports in comments 3 and 4.  We decided to use the approach which maintained a reasonable salinity difference between the Aegean and the Marmara Sea. Another possible solution is to use an Aegean Sea buffer zone to balance the Black Sea net water flux due to runoff. This means that all the extra runoff from the Black Sea would be compensated in the

Aegean. This is the strategy used by Tonani et al. (2008) for the Mediterranean Sea and tested in this model domain by Gürses et al. (2016).

**Author' response to RC#3**

**Major comments**

*However, it is necessary to discuss the result from the validation, showing that the model error increases in time (Table 2). From Figure 7 it is seen the negative trend in the volume temperature and positive for the volume salinity, what could be the reason for this trend?*

The reason for the trend is the chosen initial condition in the Aegean Sea. It takes about four years in Marmara Sea to reach statistical stability when we initialize the model from horizontally uniform initial conditions. The negative trend for temperature is connected to the heat flux forcing while the salinity to the water flux forcing.

*The section on the sea level and mass transport in the straits is also interesting. Figure 11 does not quite clear show what is the correlation between observations and simulations.*

The correlation between the observed and simulated sea level differences is computed as 0.56. We will include it in the revised manuscript.

*There is information on the lower and upper layer velocity, but it is advisable to give more details on the variability of the interface depth.*

We provided some information for the spatial variability of the interface depth in different strait junctions in Table 3 of the manuscript.

*The chapter on Marmara Sea dynamics is well written. Figure 15 and 17 might look better in color as Figure 16.*

Thanks. We prefer to keep the figures as they are since we want to put emphasis on contours and vectors.

**Some concrete remarks:**

*-The terms for α and β are thermal expansion and haline contraction coefficients, their values might be also given in the table with parameters.*

They are computed in the model following **McDougall, T. J. (1987). Neutral surfaces. Journal of Physical Oceanography,17(11), 1950-1964**.We include the reference in the revised version with an explanation.

*- The Figure 3 could be also in colors, now it is not very well read.*

Thanks, we provided a colored version of the Fig. 3.

*- Figure 6b shows cooling on the surface at the south end of Bosphorus, is it realistic?*

We think it is possible by surfacing of the cold tongue following the hydraulic jump due to the control exerted by the contraction in the Bosphorus may lead such an anomalous spots in the entrance of the Marmara Sea.

*-There is an unfinished sentence with ??????*

Thanks, the sentence is correct after removing the question marks. It is a typo after compilation by latex.

*I want to bring to the attention of authors that: a) 53 vertical levels in the Bosporus, which in some places is ∼30 m deep, is less than 1 m vertical resolution with the used vertical coordinates, that is better than the resolution used by authors. They say "The vertical resolution is 1 m in the first 50 m". The conclusion is that authors have to spend some time to reading carefully what other scientist have published. b) SCHISM uses explicit Eulerian-Lagrangian approach for momentum advection (which is unconditionally stable). It also uses an implicit scheme for terms in the momentum and continuity equations that place most stability constraints (pressure gradient, divergence, vertical viscosity). Most importantly, the size of matrix from the implicit scheme is determined by number of horizontal nodes, not this times number of levels. In fact we have used 92 levels in one version of our Kuroshio simulation and it went fine. So it's not a fundamental problem for us to use a large number of levels - it's just a practical consideration given our limited computational resource (see also my comment (a)). In any case, your results inspired us to try larger number of levels in the future.*

*Our experience with our own and other Z-coordinate models is that it requires a large number of levels to get reasonable stratification and bottom intrusion right (due to stair- case). We switched to hybrid coordinates precisely because of this. I hope that authors will avoid conveying this kind of mis-information.*

1-a) Here, Dr. Zhang mentioned our interpretation on the vertical discretization used in Stanev et al. (2017). As quoted by Dr. Zhang, we said "However, the computational burden of using an implicit scheme imposed a coarser model resolution with 53 vertical levels at the deepest point of the Black Sea". Dr. Zhang responded as "53 vertical levels in the Bosporus, which in some places is ∼30 m deep, is less than 1 m vertical resolution with the used vertical coordinates, that is better than the resolution used by authors." Here, we note that Stanev et al. (2017) states the 53 vertical levels in the deepest part of the Black Sea, not in the Bosphorus as claimed by Dr. Zhang. The exact phrase that is written in Stanev et al. (2017) is (in section 3.1 paragraph 3): "The vertical LSC2 grid consists of up to 53 levels in the deepest parts of the Black Sea, with an average number of 31.65 levels in the whole model domain." Our interpretation is that relatively shallow areas in the model domain (we assume Azov Sea, Bosphorus and other straits since it is not explicitly mentioned by the authors) should have even less

than 31.65 levels to have it as an average number. Therefore, we have difficulty to understand when Dr. Zhang claims that Stanev et al. (2017) has higher vertical resolution than our model has. We would appreciate if they could have been clearer in their paper. We kindly ask Dr. Zhang to clarify the point if we miss anything and have to modify our interpretation. Finally, we are happy that Dr. Zhang confirms us by saying "our own and other Z-coordinate models is that it requires a large number of levels to get reasonable stratification" but afraid that the mentioned requirement is not satisfied by Stanev et al. (2017), especially in the straits.

1-b) We are happy with the work presented in Stanev et al. (2017) since they show the capabilities of the unstructured meshes in modeling the Turkish Straits System. We appreciate their efforts in using implicit scheme for time discretization since explicit schemes has very strict CFL limitations as we discussed in the manuscript. However, although we don't claim of being experts on implicit schemes, we would prefer to see some discussion in Stanev et al. (2017), on the accuracy as they did on the stability since we know that accuracy of the solution can be degraded when implicit schemes are used especially in very dynamic regimes as in Bosphorus. Moreover, besides not having fully understand the relation of Stanev et al. 2017 with their work on Kuroshio current, Dr. Zhang admits the computational cost of using implicit schemes and as a result they kept the vertical resolution lower than needed in Stanev et al. (2017) as we concluded. According to us, this is a crucial mistake in modelling especially the Bosphorus Strait. Because of the same challenge, we preferred to increase the vertical resolution rather than including all the Black Sea as we proposed in our conclusions and performed by Stanev et al. (2017). Although their hybrid LSC2 grid allows to increase the resolution at the surface and the bottom, we think that they missed a lot around the interface between the upper and lower layers. The strong mixing, probably due to the diffusion, can be seen from their Fig. 5c , quite easily. If they are interested to see a better representation of the interface, they can have a look at Fig. 12 or Fig. 8 of Sannino et al. 2017 in the models and Figure 4 of Gregg et al. (1999) from the observations.

2. *I expected, after having seen their wrong criticism against the work of Stanev et al. (2017), to see some examples. Unfortunately, this paper is only cited in the introduction, so their statement there is totally unjustified. A fair approach would be to clearly demonstrate what their progress is in comparison with earlier unstructured-grid experiments.*

2) Stanev et al. (2017) focuses on the Black Sea by resolving also the Turkish Straits System. However, our main focus is the Marmara Sea which is barely analyzed in Stanev et al. (2017). Therefore, we don't see any reason to make any comparison as we did, for example, with Sannino et al. (2017) which has a focus similar to us. However, we are happy to refer Stanev et al. (2017) in the introduction being aware that studies on the Turkish Straits System attract more and more attention.

**Author' response to SC#2**

*My major concern with this study is on the capability of the present model application to correctly reproduce the water circulation in the system and the exchange dynamics at the Dardanelles and Bosphorus straits. The authors argue in the abstract and in the results description that the simulation maintains its realism. However, the evidences provided in the paper demonstrate that some numerical model results are unrealistic.*

We do not have a claim that we model every aspect of the TSS with very high accuracy. Here, what we present for the first time is a long-term simulation of the system and analysis of the evolution of the system under realistic atmosphere. Since it is done for the first time, we do not understand how Dr. Ferrarin reached a conclusion that some results are unrealistic.

*In particular: - The water fluxes presented in the table 4 are about half than what is reported in the literature (Table 5).*

We discuss the issue extensively in our response to the comment #3 of the Reviewer #2.

*- The water fluxes presented in Figure 13 do not reproduce the observed variability and magnitude.*

We discuss the issue extensively in our response to the comment #3 of the Reviewer #2.

*- The thermocline and halocline are generally deeper that what is observed (Figure 9).*

That is something that we mention when we discuss higher RMS errors around the pycnocline. Although different factors such as the insufficient variability of the Bosphorus inflow, coarse resolution of atmospheric forcing may be the reasons, we want to stress that the profiles in Fig. 9 are the mean of a couple of days and it is difficult to reproduce exactly the same profiles without any data assimilation.

*- The numerical model seams not able to correctly reproduce the observed sea level differences between Yalova and Sile. The authors should provide statistics (RMSE, BIAS, R2) of the model performance for the water levels.*

We think the model is capable of reproducing the sea level difference as much as the atmospheric forcing permits. We believe the solution will substantially improve with a higher resolution atmospheric forcing, which can only be provided by a regional model, currently.

*Concluding, to my opinion, the authors have to provide clear and robust calibration and validation of their numerical model application before inferring on the water circulation of the Turkish Strait System.*

[revised manuscript text omitted]